

# Using feature importance as exploratory data analysis tool on earth system models

Daniel Ries[1], Katherine Goode[1], Kellie McClernon[1], and Benjamin Hillman[1]

[1]Sandia National Laboratories. Albuquerque, NM. United States of America.

**Correspondence:** Daniel Ries (dries@sandia.gov)

**Abstract.**

Machine learning (ML) models are commonly used to generate predictions, but these models can also support the discovery of new science. Generating accurate predictions necessitates that a model captures the structure of the underlying data. If the structure is properly extracted, ML could be a useful exploratory and evidential tool. In this paper, we present a case study that

demonstrates the use of ML for exploratory data analysis (EDA) in the climate space. We apply the ML explainability method of spatio-temporal zeroed feature importance (stZFI) to understand how climate variable associations evolve over space and time. Our analyses focus on data from ensembles of earth systems models (ESMs), which provide data on different climate states and conditions. We elect to work with ESM ensembles since they allow us to compare feature importance across alternative scenarios not available with observed data. The ensembles also account for natural variability, so we can distinguish between

signal and noise due to natural climate variability when computing feature importance. For our analyses, we consider the 1991 volcanic eruption of Mount Pinatubo: a large stratospheric aerosol injection. We explore the climate pathway associated with the eruption from aerosols to radiation to temperature at both the near-surface and stratospheric levels. In addition to applying the method to data generated from two different ESMs, we apply stZFI to reanalysis data to compare the associations identified by stZFI. We show how stZFI tracks the importance of aerosol optical depth over time on forecasting temperatures. This case

study illustrates usefulness of an ML tool (stZFI) for EDA on a well studied climate exemplar.

## 1 Introduction

Climate science questions are often studied using ensembles of Earth Systems Models (ESMs). Since we cannot conduct global controlled climate experiments to understand cause and effect, ESMs allow climate scientists to explore the effects of different climate conditions on the climate system. However, ESMs generate large quantities of data (considering number of

ensembles, spatial resolution, temporal resolution, etc.), which can be difficult to process and understand. Therefore, methods that summarize and identify trends are valuable for working with data from ESMs. Exploratory data analysis (EDA) is the general approach to exploring, analyzing, and summarizing patterns in data. EDA includes the computation of simple summary statistics such as means, standard deviations, and correlations along with data visualizations. These approaches provide a high level view of trends but can overlook important details. More sophisticated EDA techniques allow scientists and practitioners

to understand detailed trends in the data, which promotes the ability to draw conclusions and propose new hypotheses. Our



objective in this paper is to present a case study showing the utility of a new EDA technique that leverages the data driven modeling approach of machine learning (ML) and ESMs to gain insights into climate problems.

ESMs provide a mathematical representation of the complex and chaotic nature of Earth's climate. When these models are run with different initial states, parameter values, external forcings, etc., the models produce an ensemble of simulations that represent different possible climate scenarios. The ensembles provide an estimate of model and natural variability given a particular forcing and enable investigation of possible climate outcomes that are not realized in the observational record. From a ML perspective, these ensembles represent a set of temporal "replicates" where the true underlying relationships are known. In contrast, observational climate data only provide a single instance in space-time. Observed data are also limited to events in the past, which may not include all events that researchers and policy-makers are interested in studying. For example, no major stratospheric aerosol injection experiment has been conducted in the field, but we are still interested in the impact such scenario would produce. ESMs provide a way to understand variable relationships not seen in observational data and capture natural climate variability.

There is a rich history of using ML and statistical models for analyses with ensembles of ESMs. Examples of these analyses include Tebaldi et al. (2005) and Smith et al. (2009) who built statistical models to quantify the uncertainty of replicates from different ESMs. Going the other direction, many approaches have been developed which use observational data to calibrate climate model replicates (e.g., Reichler and Kim (2008), Armour et al. (2013), and Baker et al. (2016)). More recently, ML models are used to provide insight into climate processes by quantifying relationships between ESM variables (e.g., Hart et al. (2023) and de Burgh-Day and Leeuwenburg (2023)). McClernon et al. (2024) considered the assessment of ML models used in such situations and developed a cross-validation procedure using ESM ensembles to obtain a true *replicate hold-out* set to assess the commonly used *repeated hold-out* cross-validation process for time series data (Cerqueira et al., 2020). .

While ML models are commonly used for prediction applications, their ability goes beyond simple prediction. Data driven models are capable of both finding new patterns and verifying known relationships. As Toms et al. (2020) points out, "the ultimate objective of using a neural network can also be the interpretation of what the network has learned rather than the output itself". However, many ML models are black box algorithms whose mathematical formulas are too complex to interpret variable relationships captured by the model. Explainability methods applied to ML models provide a link from the predictive power of the ML model to an understanding of the underlying processes. Goode et al. (2024) defined a model as being *explainable* if it is possible to implement post hoc investigations on a trained model that infer how the model inputs relate to the model outputs.

The climate science community has recently recognized the utility of explainable ML methods. To find variables that best discovered model errors in an ESM, Silva et al. (2022) use an explainable ML method, SHapley Additive exPlanation (SHAP) values. Toms et al. (2020) use backward optimization and layerwise relevance propagation to discover scientifically meaningful connections with respect to ENSO phase detection and prediction. McGovern et al. (2019) provide an overview of potential explainability for ML applied to meteorology. Clare et al. (2022) apply Bayesian neural networks with both layer-wise Relevance Propagation (LRP) and SHAP values to better characterize and quantify ocean circulation dynamics. On the ESMs ARISE-SAI Mamalakis et al. (2023) explore the impacts of stratospheric aerosol injections on different variables with he explainability method Deep SHAP (Lundberg and Lee, 2017).





Explainability techniques possess potential for providing insight into patterns in data captured by a black-box model, but research has also identified pitfalls with current methods (e.g., Rudin (2019); Hooker et al. (2021); Ancona et al. (2017)). Mamalakis et al. (2022) compares different convolutional neural network explainability methods by utilizing synthetic data so the "true explanations" are known *a priori*. Their analysis highlights the strengths and weaknesses of various methods, and

they conclude "that no optimal method exists for all prediction settings". They recommend applying and comparing results from various explainability methods while more rigorous assessments of explainability techniques are needed. We believe the case study in this paper will contribute to this body of understanding.

In this paper, we present a case study that demonstrates the explainability technique of spatio-temporal zeroed feature importance (stZFI; Goode et al., 2024) as an EDA tool for a climate problem that leverages earth system model (ESM) ensembles.

Goode et al. (2024) developed stZFI for echo state networks (ESNs), which are a computationally fast, yet powerful, ML model for spatio-temporal data. stZFI measures the relative gain in predictive performance for each input, or predictor, variable over time. This allows users to see how "important" input variables are for the predictive ability of the ML model and provides insight into the dynamic nature of the relationships. We use stZFI to explore relationships between pathway variables associated with a stratospheric aerosol injection climate event. We apply stZFI to ensembles of ESMs, which allows us to measure uncer-

tainties in variable importances that effectively account for variability. Our analyses are intended to showcase the applicability of stZFI as an exploratory and evidential tool for climate related problems.

## 1.1 Motivating Application

Stratospheric aerosol injection (SAI) is being studied as a possible way of mitigating climate change (Irvine et al., 2016), but there is concern over its potential side effects (MacMartin et al., 2016; McCormack et al., 2016). Although there is an abundance

of computer model experiments looking at SAI (Ferraro et al., 2015; Banerjee et al., 2021; Bednarz et al., 2022), we are unaware of any physical SAI experiments. In lieu of SAI experiments, the 1991 eruption of Mount Pinatubo provides a natural exemplar of a large SAI event. The eruption released 18-19 Tg of sulfur dioxide into the atmosphere, causing changes to aerosol optical depth (AOD) and consequently to stratospheric temperatures (Sato et al., 1993; Guo et al., 2004). The increase in AOD scatters shortwave radiation (Twomey, 1991) and absorbs and re-emits longwave radiation (Zhou and Savijärvi, 2014). The increase in

shortwave scattering tends to cool the earth surface by reflecting more incoming solar radiation, while the increase in longwave absorption tends to warm the lower stratosphere. As a consequence of Mount Pinatubo's eruption, Temperatures at pressure levels of 30 to 50 mb rose between 2.5 to 3.5 degrees centigrade (Labitzke and McCormick, 1992), while temperatures at the surface decreased by 0.5 degrees centigrade (Parker et al., 1996).

We purposely examine the well studied eruption of Mount Pinatubo since our goal is to demonstrate the usefulness of stZFI

as an EDA tool. By picking a well known event and phenomenon, we can compare relationships identified by stZFI with previously identified relationships in the scientific literature. An agreement in identified relationships could provide confidence in the proposed approach. We apply stZFI to data generated from a simplified ESM and a fully coupled (i.e., allowing for interactions between atmosphere, land, ocean, etc.) ESM. We additionally consider one reanalysis dataset (i.e., combination of observed and model data) to demonstrate the ability of stZFI to find interesting relationships in observations and quantify





how they evolve over time. When used as an EDA tool, these analyses show how stZFI can be a useful way to understand the complex relationships in the data.

The remainder of this article is structured as follows. Section 2 introduces the ML model and explainability method used in this paper, the ensemble echo state network (EESN) and stZFI, respectively. Section 3 introduces the data sets: HSW++ (Hollowed et al., 2024b), Energy Exascale Earth System Model (E3SM), (Rasch et al., 2019; Golaz et al., 2022), and the Modern-Era Retrospective Analysis for Research and Applications, Version 2 (MERRA-2) (Gelaro et al., 2017) reanalysis. This section additionally quantifies the variable relationship using stZFI for each data set. Section 4 makes comparisons between E3SM and MERRA-2 results. Finally, Section 5 discusses results, conclusions, and future directions.

## 2 Data Model

This section reviews the EESN and stZFI approaches used to measure climate variable relationships. The EESN and stZFI assume data is centered and scaled prior to model training to improve model performance and make interpretations of importances easier. The data throughout this section is assumed to be centered and scaled according to a preprocessing procedure of the modeler's choice.

### 2.1 Ensembled Echo State Network

Echo state networks (ESNs) (Jaeger, 2001; Lukoševičius and Jaeger, 2009) are known to provide good predictions for chaotic systems (Alao et al., 2021). ESNs are also computationally efficient in comparison to recurrent neural networks, their sibling ML model for temporal data. The ESN applied to spatio-temporal climate data was first explored by McDermott and Wikle (2017) and improved upon in McDermott and Wikle (2019). We follow the notation of Goode et al. (2024) and McClernon et al. (2024), since it allows for an easier presentation of feature importance (FI) in the next section. Let:

$$\mathbf{Z}_{Y,t} = (Z_{Y,t}(\mathbf{s}_1), Z_{Y,t}(\mathbf{s}_2), ..., Z_{Y,t}(\mathbf{s}_N))', \tag{1}$$

be the vector of preprocessed responses at locations $\{\mathbf{s}_i \in \mathcal{D} \subset \mathbb{R}^2; i = 1, ..., N\}$ over times $t = 1, ..., T$. The preprocessed model inputs are also spatio-temporal processes, represented as:

$$\mathbf{Z}_{k,t} = (Z_{k,t}(\mathbf{s}_1), Z_{k,t}(\mathbf{s}_2), ..., Z_{k,t}(\mathbf{s}_N))', \tag{2}$$

$k = 1, ..., K$. The locations of all variables are assumed to be the same, which is expected for climate model simulations. To reduce spatial dimensionality, we use empirical orthogonal functions (EOFs) to decompose the variable anomalies such that:

$$\mathbf{Z}_{Y,t} \approx \mathbf{\Phi}_Y \mathbf{y}_t \tag{3}$$

$$\mathbf{Z}_{k,t} \approx \mathbf{\Phi}_k \mathbf{x}_{k,t}, \tag{4}$$

for $k = 1, ..., K$, where $\mathbf{\Phi}_Y$ is an $N \times Q$ matrix of EOFs corresponding to $\mathbf{Z}_{Y,t}$ and $\mathbf{\Phi}_k$ is an $N \times P_k$ matrix of EOFs corresponding to $\mathbf{Z}_{k,t}$. $\mathbf{y}_t$ and $\mathbf{x}_{k,t}$ are vectors of length $Q$ and $P_k$, respectively, which are the scores from the EOF decomposition.





$Q$ and $P_k$ are user chosen hyperparameters corresponding to the number of EOFs for the output and $k^{th}$ input, respectively. These values can be chosen using hyperparameter tuning or considering computational complexity. without loss of generality, we will assume $P_1 = P_2 = ... = P_k = P$ throughout.

McDermott and Wikle (2019) introduced the EESN as a way to quantify uncertainty and improve predictions by averaging over different initializations of the reservoir. The EESN is given by:

$$\textit{Output stage:} \quad \mathbf{y}_t = \mathbf{V}^{(r)}\mathbf{h}_t + \boldsymbol{\epsilon}_t^{(r)} \tag{5}$$

$$\textit{Hidden stage:} \quad \mathbf{h}_t = g_h\left(\frac{\nu}{|\lambda_w|}\mathbf{W}^{(r)}\mathbf{h}_{t-\tau-\tau^*} + \mathbf{U}^{(r)}\tilde{\mathbf{x}}_{t-\tau}\right), \tag{6}$$

$$\textit{Regression Error:} \quad \boldsymbol{\epsilon}_t^{(r)} \sim Gaussian\left(\mathbf{0}, \sigma_\epsilon^{2(r)}\mathbf{I}\right), \tag{7}$$

where $r = 1, 2, ..., R$ represents the EESN ensemble member. We omit the quadratic term included by McDermott and Wikle (2019). During our initial tuning of the models, we found increasing the size of the EESN was more beneficial than including additional terms, although this could be treated as part of model selection. Input variables are included in the embedding vector, $\tilde{\mathbf{x}}_{t-\tau}$, which is defined by

$$\tilde{\mathbf{x}}_{t-\tau} = \left[\mathbf{x}'_{t-\tau}, \mathbf{x}'_{t-\tau-\tau^*}, ..., \mathbf{x}'_{t-\tau-m\tau^*}\right]'. \tag{8}$$

$\tau$ is the forecast period (i.e., how many steps ahead in time the EESN will make predictions) and should be chosen based on the goals of the modeler. $\tau^*$ is the embedding vector lag, and $m$ is the number of embedding lags. Both are pre-specified and can be selected either during hyperparameter tuning or based on subject matter expertise.

$\mathbf{h}_t$ contains the $n_h$ *hidden units* which include information on the inputs beyond the immediate past, where $n_h$ is a tuning parameter. The matrices $\mathbf{W}^{(r)}$ and $\mathbf{U}^{(r)}$ contain the reservoir weights with dimensions of $n_h \times n_h$ and $n_h \times P(m+1)$, respectively. $\mathbf{W}^{(r)}$ and $\mathbf{U}^{(r)}$ are not estimated, but rather are randomly sampled R times from their respective distributions as follows:

$$\mathbf{W}^{(r)}[h, c_w] = \gamma_{h,c_w}^w \text{Unif}(-a_w, a_w) + (1 - \gamma_{h,c_w}^w)\delta_0, \tag{9}$$

$$\mathbf{U}^{(r)}[h, c_u] = \gamma_{h,c_u}^u \text{Unif}(-a_u, a_u) + (1 - \gamma_{h,c_u}^u)\delta_0, \tag{10}$$

where $\mathbf{W}^{(r)}[h, c_w]$ represents the element row $h$ and column $c_w$ of $\mathbf{W}^{(r)}$, and similarly, $\mathbf{U}^{(r)}[h, c_u]$ represents the element in row $h$ and column $c_u$ of $\mathbf{U}^{(r)}$. $\gamma_{h,c_w}^w \sim Bern(\pi_w)$, $\gamma_{h,c_u}^u \sim Bern(\pi_u)$, and $\delta_0$ is a Dirac function. Sampling multiple times from reservoir distributions allows us to have a distribution of predictions over which to average and calculate uncertainty. $a_w$, $a_u$, $\pi_w$, and $\pi_u$ serve as regularization hyperparameters to prevent overfitting. $\nu$ is a value in $[0, 1]$ that helps control the amount of memory in the system through $\mathbf{h}_t$. $\lambda_w$ is the spectral radius of $\mathbf{W}$. $g_h$ is a nonlinear activation function for which we use a hyperbolic tangent function. The only parameters estimated in the model are contained in the matrix $\mathbf{V}^{(r)}$ and the error term $\sigma_\epsilon^{2,(r)}$. $\mathbf{V}^{(r)}$ is a $Q \times n_h$ parameter matrix of coefficients estimated using a ridge regression with a penalty parameter of $\lambda_r$. This ridge regression adds another layer of regularization to the model to prevent overfitting. Lukoševičius (2012) and Goode et al. (2024) provide recommendations and results for tuning ESNs.





## 2.2 Feature Importance

Goode et al. (2024) introduced stZFI as a FI metric for assessing variable importance and its evolution over time for spatio-temporal data. Feature importance is a quantitative measure of how important an input variable is for accurately predicting an output variable at a particular time. stZFI provides a quantitative measure of importance for an input variable over time by measuring the increase in a predictive metric when the variable is removed at each time. Goode et al. (2024) computed stZFI for individual ESNs. We adjust the approach for an ensemble of ESNs. In particular, we compute stZFI using the ensemble prediction (i.e., the average of the predictions produced by each ensemble member in the EESN). This is in contrast to an approach that computes stZFI for each member of the EESN and average the stZFI results across ensembles, which is less in line with how EESNs would be used in practice to obtain the "final" prediction.

### 2.2.1 stZFI Global Metric

stZFI measures the importance of input variables, over a block of times, $\{t, t-1, ..., t-b+1\}$, $b \in \mathbb{N}$, on the forecasts of the spatio-temporal output variable, at time $t+\tau$, averaged over locations. To simplify notation, and without loss of generality, we assume $\tau^* = 1$. Let $f^{(r)}(\mathbf{x}_t, \mathbf{x}_{t-1}, ..., \mathbf{x}_1) = \hat{\mathbf{y}}_{t+\tau}^{(r)}$ represent the vector of forecasts from the $rth$ ensemble of the EESN, at time $t+\tau$ given $\mathbf{x}_t, \mathbf{x}_{t-1}, ..., \mathbf{x}_1$, and let $\bar{\hat{\mathbf{y}}}_{t+\tau} = \frac{1}{R}\sum_{r=1}^{R}\hat{\mathbf{y}}_{t+\tau}^{(r)}$ be the aggregated forecast from the EESN. Define the root mean squared error (RMSE) on the spatial scale as

$$RMSE_{t+\tau} = Q^{-1/2}\|\mathbf{Z}_{Y,t+\tau} - \mathbf{\Phi}_Y \bar{\hat{\mathbf{y}}}_{t+\tau}\|. \tag{11}$$

The procedure for stZFI also computes an "zeroed" RMSE, $RMSE_{t+\tau}^{*(k)}$, that captures model predictive performance when zeroing all EOF scores for an input feature $k$. First, replace the vectors of $\mathbf{x}_{k,t}, \mathbf{x}_{k,t-1}, ..., \mathbf{x}_{k,t-b+1}$ within $\mathbf{x}_t, \mathbf{x}_{t-1}, ..., \mathbf{x}_{t-b+1}$ with zeros. Denote these as $\mathbf{x}_t^{(k)}, \mathbf{x}_{t-1}^{(k)}, ..., \mathbf{x}_{t-b+1}^{(k)}$, respectively. Next, compute forecasts using the zeroed inputs as

$$f^{(r)}\left(\mathbf{x}_t^{(k)}, \mathbf{x}_{t-1}^{(k)}, ...\mathbf{x}_{t-b+1}^{(k)}, \mathbf{x}_{t-b}, ..., \mathbf{x}_1\right) = \hat{\mathbf{y}}_{t+\tau}^{(r,k,b)}. \tag{12}$$

Similarly, let $\bar{\hat{\mathbf{y}}}_{t+\tau}^{(k,b)} = \frac{1}{R}\sum_{r=1}^{R}\hat{\mathbf{y}}_{t+\tau}^{(r,k,b)}$ be the aggregate forecast from the EESN when variable $k$ is zeroed at time $t$ with block size $b$. The RMSE after zeroing relevant variable $k$ EOF scores on the spatial scale is:

$$RMSE_{t+\tau}^{*(k)} = Q^{-1/2}\|\mathbf{Z}_{Y,t+\tau} - \mathbf{\Phi}_Y \bar{\hat{\mathbf{y}}}_{t+\tau}^{(k,b)}\|, \tag{13}$$

The stZFI metric for forecasting time $t+\tau$, for variable $k$, and block size $b$ is computed as:

$$\mathcal{I}_{t,t+\tau}^{(k,b)} = RMSE_{t+\tau}^{*(k)} - RMSE_{t+\tau}. \tag{14}$$

Larger values of stZFI mean variable $k$ is relatively more important to the model for making predictions, and therefore important for describing the pathway from input to output. Values of stZFI near zero imply the variable has little impact on predictions, and therefore, does not show a strong association with the output variable. The values of the feature importance metric itself are reductions in predictive RMSE and can be interpreted as such. For example, a feature importance of $\mathcal{I}_{t,t+\tau}^{(k,b)} = 2$ for variable $k$ at time $t$ means the RMSE increases by 2 units with variable $k$ at time $t$ removed from the model.





### 2.2.2 stZFI Regional Metric

The stZFI metric in Equation (14) is a *global* metric; it measures the impact an input variable has on an output variable, on a globally averaged scale. The metric can be decomposed regionally by calculating the contributions to $\mathcal{I}_{t,t+\tau}^{(k,b)}$ by spatial regions such as latitude bands. Regional contributions to stZFI provide the ability to quantify the impact of a global input variable on regional output variable. In this paper, we only consider latitude bands for regional contributions to stZFI, so we incorporate this in our notation, but more generally, other spatial regions could be considered.

Let the regional feature importance metric be represented by:

$$\mathcal{I}_{t,t+\tau}^{(k,b)}[lat] = RMSE_{t+\tau}^{*(k)}[lat] - RMSE_{t+\tau}[lat]. \tag{15}$$

where $lat$ represents all locations $\mathbf{s}_i$ in the latitude band $lat$ for output variables where $lat$ indicates the average measure across all locations $\mathbf{s}_i$ in the defined latitude band. The first RMSE on the right hand side of Equation 15 is the RMSE for the latitude band $lat$, when variable $k$ is zeroed globally:

$$RMSE_{t+\tau}^{*(k)}[lat] = Q^{-1/2}\|\mathbf{Z}_{Y,t+\tau}[lat] - \mathbf{\Phi}_Y \bar{\hat{\mathbf{y}}}_{t+\tau}^{(k,b)}[lat]\|, \tag{16}$$

The second RMSE in Equation 15 is the RMSE for all locations $\mathbf{s}_i$ with latitudes equal to $lat$, much like the RMSE in Equation 11, except it only considers data with latitude equal to $lat$:

$$RMSE_{t+\tau}[lat] = Q^{-1/2}\|\mathbf{Z}_{Y,t+\tau}[lat] - \mathbf{\Phi}_Y \bar{\hat{\mathbf{y}}}_{t+\tau}[lat]\|. \tag{17}$$

## 3 Stratospheric Aerosol Injection Applications

We apply stZFI to data from two climate models (HSW++ and E3SM; Sections 3.1 and 3.2, respectively) and one reanalysis dataset (MERRA2; Section 3.3). We consider these three data sources in order to compare the behavior of feature importance across different techniques for data acquisition associated with the same SAI climate event. HSW++ is a simplified climate model while E3SM is a fully-coupled model. Both HSW++ and E3SM have counterfactual runs, which allow us to compute stZFI when no major injection of aerosols occurs. MERRA-2 gives us a representation of the observed climate and allows us to compare real values to values generated by ESMs. Details on each dataset, data preprocessing, and FI results are presented in this section.

First, standardization is performed to ensure all input variables in the EESN are on the same scale such that feature importances are comparable between variables. Specifics for each dataset are described within each data's subsection. For ease of illustration and comparison, we trained EESNs on all datasets with the same $\tau, \tau^*, m$ and $b$ values. We use $\tau = 1$ since we are interested in relatively short-lead forecasting, and we set $m = 3$ and $\tau^* = 1$ as an example of a model where emphasis is placed on the past quarter of a year for a prediction. For stZFI block size, we elect to use $b = 4$ since with HSW++, E3SM, and MERRA-2 data for the Pinatubo eruption application, we have found block sizes larger than this often do not change results much, potentially indicating sufficient removal of auto-correlation in importance. Several other values were kept constant





across data sets. We set $R = 10$ and use the first 20 EOFs from each climate variable for training the EESN. We will use $R = 10$ ensemble members for all EESNs to balance computational complexity of stZFI and predictive performance. We use 20 EOFs for all variables in this paper to keep comparisons simple and fair. We select 20 EOFs for computational convenience, but this value could also be tuned as part of hyperparameter optimization. The remaining EESN hyperparameters were optimized using a grid search. The procedure was implemented separately for each dataset. The data were split into training and test sets, and

the hyperparameter set giving the lowest test set predictive performance was used to compute the feature importances presented in this section. Details on the EESN hyperparameter selection and tuning process is in Appendix A. A predictive assessment of the EESNs for each dataset with their best performing hyperparameters is provided in Appendix B.

## 3.1 HSW++

We first consider a simplified ESM with a single stratospheric injection of aerosols referred to as *HSW++* (Hollowed et al.,
2024b). The primary use for this simplified climate model is to verify that stZFI finds relationships built into the ESM that are less likely to be entangled with higher order effects. This makes it easier to verify how the method behaves in an intuitive and predictable scenario. Held and Suarez (1994) created an idealized forcing without topography and seasonality; this is combined with the modified temperature equation of Williamson et al. (1998), which allows for modeling stratospheric temperatures which are necessary when considering an SAI. HSW++ further modifies the temperature equation by making adjustments
based on the observed aerosols at a given pressure level. HSW++ runs approximately 168 times faster than E3SM (McClernon et al., 2024), making it useful for initial model evaluations.

The aerosol injection is meant to resemble Mount Pinatubo's eruption in size, space, and time. Model outputs are remapped to a $2° \times 2°$ structured latitude/longitude grid with 72 vertical levels. The temporal resolution of the output is 48 hours. The simulations begin at day 0 and run for 1200 days, with the injection on day 179. There is no seasonality in the model, and
the background radiation is prescribed to be in balance. Aerosols come only from the single Pinatubo-like injection of sulfate precursor and volcanic ash, meaning AOD is fully driven by the prescribed injection. Surface and stratospheric temperatures are then parameterized through AOD. We build two models predicting temperature for HSW++:

– *HSW++ Stratosphere Model*: Predict T050 (temperature at 50 mb) given AOD and T050.

– *HSW++ Surface Model*: Predict T1000 (temperature at 1000 mb) given AOD and T1000.

All input variables are time-lagged. An ensemble of simulations with the HSW++ configuration are used here, with 5 ensemble members (each with perturbed initial conditions) with Pinatubo forcing and a single counterfactual (without Pinatubo) simulation. We note that McClernon et al. (2024) also explored fitting an EESN on the HSW++ data to forecast temperatures, but the model in McClernon et al. (2024) used AOD, T050, and T1000 to predict T1000.





### 3.1.1 Normalized Anomalies

Let $Z_{k,t}^O(\mathbf{s}_i)$ be the measured value for variable $k = 1, 2$, at time $t = 1, 2, ..., T$, for location $\mathbf{s}_i$, $i = 1, 2, ..., N$. The HSW++ normalized anomaly for each HSW++ ensemble for variable $k$, time $t$, location $\mathbf{s}_i$, denoted $Z_{k,t}(\mathbf{s}_i)$, is calculated by:

$$Z_{k,t}(\mathbf{s}_i) = \frac{Z_{k,t}^O(\mathbf{s}_i) - \bar{Z}_k^{CF}(\mathbf{s}_i)}{sd(\mathbf{Z}_k^{CF}(\mathbf{s}_i))}. \tag{18}$$

where $\bar{Z}_k^{CF}(\mathbf{s}_i)$ and $sd(\mathbf{Z}_k^{CF}(\mathbf{s}_i))$ are the mean and standard deviation, respectively, computed from the counterfactual run across all times for variable $k$ at location $\mathbf{s}_i$. Normalized anomalies for the temperature response, $\mathbf{Z}_{Y,t}(\mathbf{s}_i)$, are calculated

similarly. For HSW++, $k = 1$ refers to AOD and $k = 2$ refers to T050 or T1000, depending on which model is being discussed. The counterfactual in HSW++ has AOD equal to zero for all times and locations, so we set $\bar{Z}_1^{CF}(\mathbf{s}_i) = 0 \ \forall \ i$, and instead of $sd(\mathbf{Z}_1^{CF}(\mathbf{s}_i))$, we calculate $sd(\mathbf{Z}_1^O(\mathbf{s}_i))$, the standard deviation across measured AOD. Because AOD is zero for all times and locations for the counterfactual, we replace it with random realizations from a standard Gaussian distribution (mean zero, standard deviation one) to correspond to normalized data.

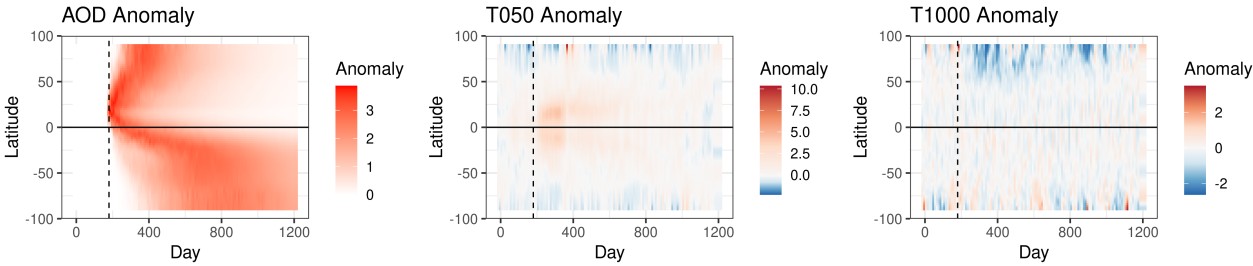

**Figure 1.** Latitudinal means over time for HSW++ Ensemble 1 normalized anomalies. Vertical dashed lines on day 179 denote the aerosol injection.

Figure 1 shows latitudinal means of the anomalies for all variables, over time, for HSW++ ensemble member 1. The injection and spread of aerosols is clear in latitude and time. Figure 2 shows globally averaged normalized anomalies for HSW++ for both the ensemble and the single-run counterfactual. The shaded regions in this figure, and throughout the remainder of the paper, represent $\pm 1$ standard deviation across model ensembles. The injection of aerosols at time $t = 179$ and how it propagates in space and time is clearly seen in both figures. T050, which is directly related to AOD in HSW++, has a strong and prompt

reaction to the injection. The response of T1000 to the injection is more noisy, but still apparent. In the counterfactual, there is a small spike in T050 around day 270 that is unrelated to an aerosol injection (counterfactual AOD is zero).

### 3.1.2 Feature Importance

After hyperparameter selection, the HSW++ models were trained using all 1200 days, and stZFI was computed. Figure 3 shows results from applying stZFI to HSW++ (using the methodology described in Section 2.2.1). The top plot shows stZFI for the

model predicting T050. The bottom plot shows stZFI for the model predicting T1000. The vertical dashed line denotes the



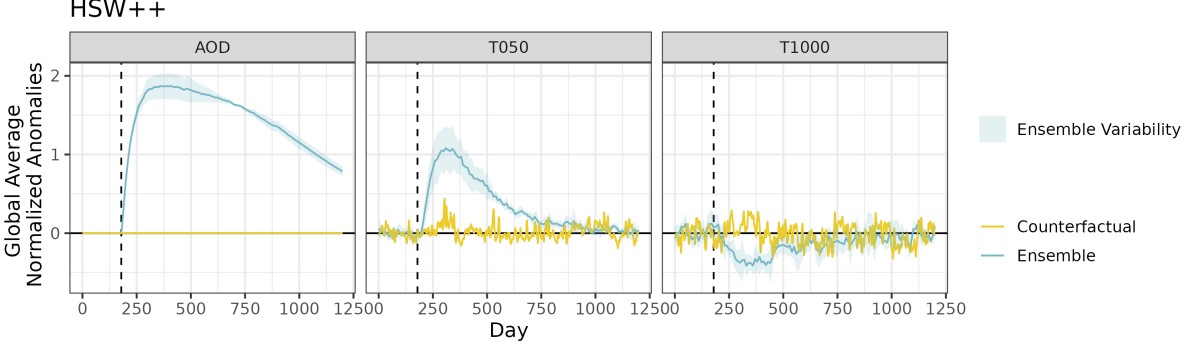

**Figure 2.** Globally averaged normalized anomalies for HSW++ for aerosol injection ensemble and counterfactual. Solid line is mean across ensembles. Vertical dashed lines on day 179 denote the aerosol injection. Shaded regions represent ± 1 standard deviation of ensemble variability. There is only one counterfactual run.

injection time. Note that the y-axis scale is different for the two plots. The importance of AOD increases sharply for the T050 at the time of injection and slowly decays thereafter, as expected. Time-lagged T050 follows a similar trend. The increased importance for AOD when forecasting T1000 is less pronounced, though present. The importance for AOD is higher for the T050 model compared to T1000 model, and the decay is smoother for the T050 model compared to the T1000 model. This

makes sense since T1000 is noisier than T050, and the true relationship between AOD and T050 is stronger than AOD with T1000. Thus, the FI results agree with our expectation that the impact AOD has on T1000 is less pronounced compared to T050.

The counterfactual run allows us to consider how stZFI responds when there is no aerosol injection. The yellow lines in Figure 3 show stZFI for the counterfactual of HSW++. Feature importance for AOD is relatively flat for both models when

considering the variation over time, which provides evidence that the peaks in stZFI for the runs with an aerosol injection are due to the EESN making use of the increase in aerosols for predicting temperatures. For the T050 model counterfactual, there is a small decline in AOD importance after the injection. This likely corresponds to the small spike in T050 previously identified in Figure 2 that is known to be unrelated to the aerosol injection.

Figure 4 shows the latitudinal contributions to stZFI for the two predictive models on HSW++ (as described in Section

2.2.2). The T050 model shows the impact of AOD on T050 after the aerosol injection around the equator. The importance of AOD lasts longer in the northern hemisphere than the southern. This is interesting since in Figure 1, the AOD anomalies are higher for a longer period of time in the southern hemisphere. Thus, even though the southern hemisphere sees higher AOD anomalies longer than the northern hemisphere, they are not identified as important for predicting T050 at the same location. Lagged T050 are important around the equator after the eruption, which matches the increased anomalies in Figure 1. For the

model predicting T1000, AOD is most important at the high northern latitudes. This is also where we see the largest negative anomalies in T1000 in Figure 1 (however, recall that importance doesn't indicate sign of relationship).



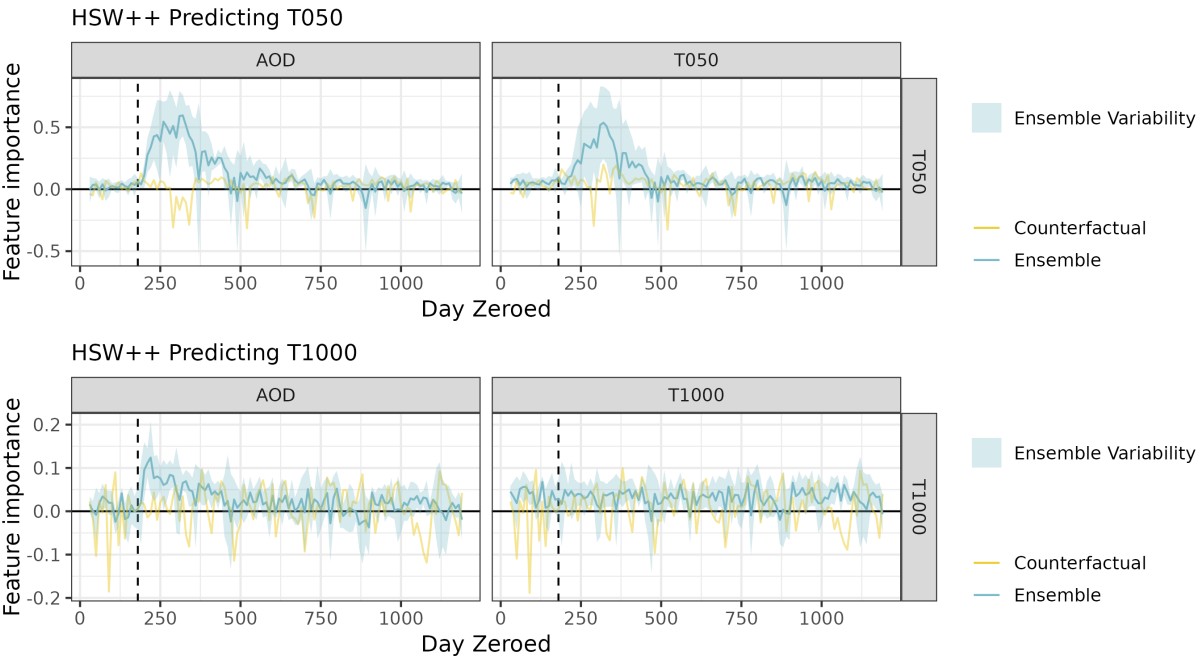

**Figure 3.** stZFI for HSW++. Vertical dashed lines denote the injection time. Shaded regions represent $\pm$ 1 standard deviation of ensemble variability.

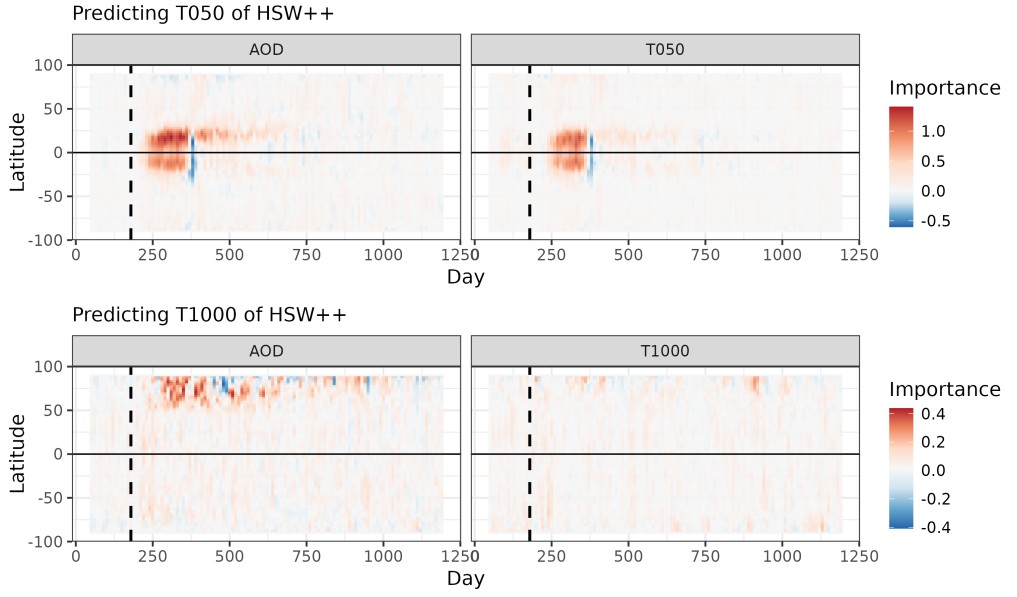

**Figure 4.** Latitudinal contributions to stZFI for models fit to HSW++. Vertical dashed lines on day 179 denote the aerosol injection.





By applying stZFI to HSW++ ensembles, we were able to assess the importance of each variable to the model in terms of predictive ability. For example, when predicting T050, we identified that AOD quickly increases in importance after the aerosol injection and then slowly decays, and stZFI is near 0 when AOD is near zero. From an EDA perspective, this suggests that an SAI event is at least associated with changes in T050. Insights such as this could inspire additional hypotheses to explore and additional settings on the ESM to run. For example, given we see this effect for a Pinatubo-like eruption, how would the impact differ if we changed the latitude of the aerosol injection? This simple example is merely a proof of concept since the injection of aerosols was the only change in the system. Next, we consider the fully coupled E3SM where the relationships resulting from Mount Pinatubo are not as obvious.

## 3.2 E3SM

E3SM is a fully-coupled state-of-the-science ESM capable of simulation and prediction created by the United States Department of Energy and its national laboratories (Rasch et al., 2019; Golaz et al., 2022). E3SM is a full-physics model with active model components consisting of atmosphere, land, ocean, sea ice, and river. Data is mapped to a $1° \times 1°$ structured latitude/longitude grid for 72 vertical levels.

The Mount Pinatubo eruption in the model occurs on June 15, 1991 at 15.14167°N and 120.35000°E. The magnitude is 10 Tg of $SO_2$ spread evenly over 6 hours at an altitude of 18-20 km. For our analyses, we consider data on the monthly time scale. Five ensembles were generated from the model. Each ensemble was initialized with perturbed initial states beginning on January 1, 1985 to ensure that by June 15, 1991, all ensembles are dynamically independent (Brown et al., 2024). Dynamics arise from seasonal heating imbalance and additional forcings beyond Mount Pinatubo that change global radiation balance. In addition, there is a positive trending background imbalance due to anthropogenic emissions of greenhouse gases. There are three aerosol precursor gases and seven aerosol species from both natural and anthropogenic sources which vary seasonally. Two meter surface temperature depends on the solar heating rate, which is affected by AOD, cloud cover, surface albedo, and ocean state. Counterfactuals with the Mount Pinatubo eruption removed were generated for each of the five ensemble members.

Again, we build two EESN models for modeling temperature pathways with the E3SM ensembles.

- *E3SM Stratosphere Model*: Predict T050 given AOD, long-wave radiative flux net top of atmosphere (LWTUP) and T050.

- *E3SM Surface Model*: Predict T2M (two meter surface temperature) given AOD, short-wave radiative flux clear sky (SWGDNCLR) and T2M.

All input variables are time-lagged. These variables form the structure explained in McCormick et al. (1995), where the Mount Pinatubo eruption injected aerosols which warmed the stratosphere with upwelling radiation and a cooling of the surface.



### 3.2.1 Normalized Anomalies

Normalized anomalies for E3SM are calculated slightly differently than HSW++ since E3SM has seasonality. The normalized anomaly for each E3SM ensemble for variable $k$, time $t$, and location $\mathbf{s}_i$ is calculated by:

$$Z_{k,t}(\mathbf{s}_i) = \frac{Z_{k,t}^O(\mathbf{s}_i) - \bar{Z}_{k,month(t)}^{CF}(\mathbf{s}_i)}{sd(\mathbf{Z}_{k,month(t)}^{CF}(\mathbf{s}_i))}, \tag{19}$$

where $\bar{Z}_{k,month(t)}^{CF}(\mathbf{s}_i)$ and $sd(\mathbf{Z}_{k,month(t)}^{CF}(\mathbf{s}_i))$ are the mean and standard deviation, respectively, computed from an ensemble's corresponding counterfactual run across all data in the month for which time $t$ belongs (i.e., $month(t)$ returns the month of time $t$), for variable $k$ and location $\mathbf{s}_i$. Normalized anomalies for the temperature response, $\mathbf{Z}_{Y,t}(\mathbf{s}_i)$, are calculated similarly. For E3SM, $k = 1$ refers to AOD, $k = 2$ refers to long-wave radiation net top of stratosphere (LWTUP) for the T050 stratospheric model or incoming radiation at surface without clouds (SWGDNCLR) for the surface (T2M) model and $k = 3$ refers to the respective temperature for each model. Figure 5 shows latitudinal means of normalized anomalies over time from a single ensemble of E3SM.

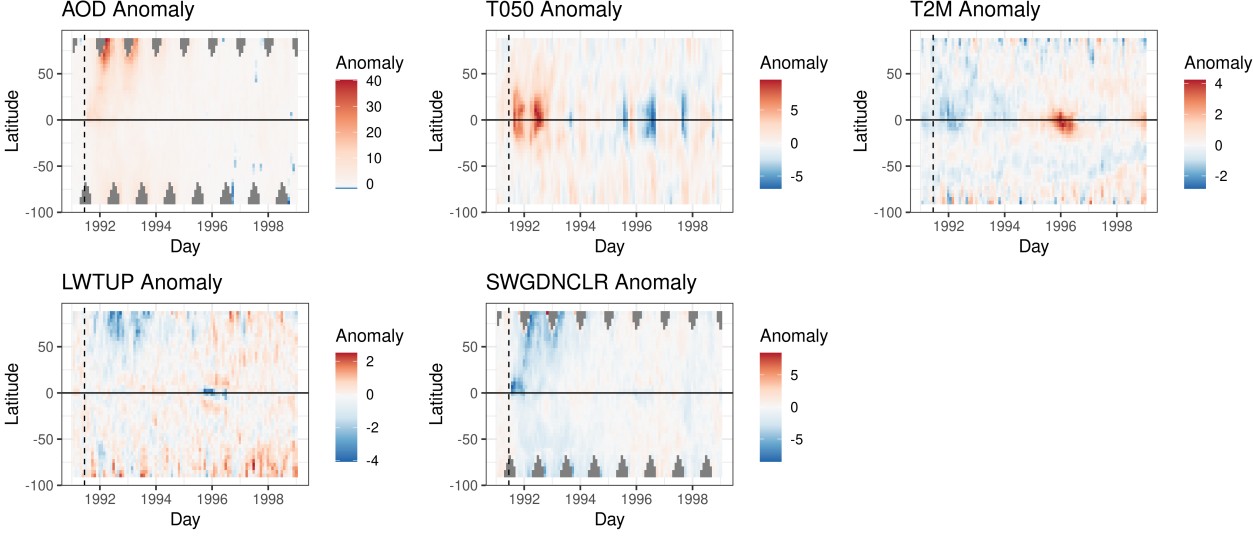

**Figure 5.** Latitudinal means over time for E3SM Ensemble 1 normalized anomalies. Vertical dashed lines denote the June 15, 1991 Mount Pinatubo eruption. Gray in AOD and SWGDNCLR plots are NA values.

Figure 6 shows globally averaged normalized anomalies for E3SM. AOD has the largest spike relative to the counterfactuals, while T2M sees the smallest change. The impact of Mount Pinatubo is clear on the radiation measurements and T050. For the counterfactual case, there is still a small spike in AOD at the end of 1991 along with small impacts to radiation and temperature even with Mount Pinatubo removed. The cause of this signal could be the volcanic eruption of Cerro Hudson on on August 8, 1991 which was smaller than Mount Pinatubo Miles et al. (2017).





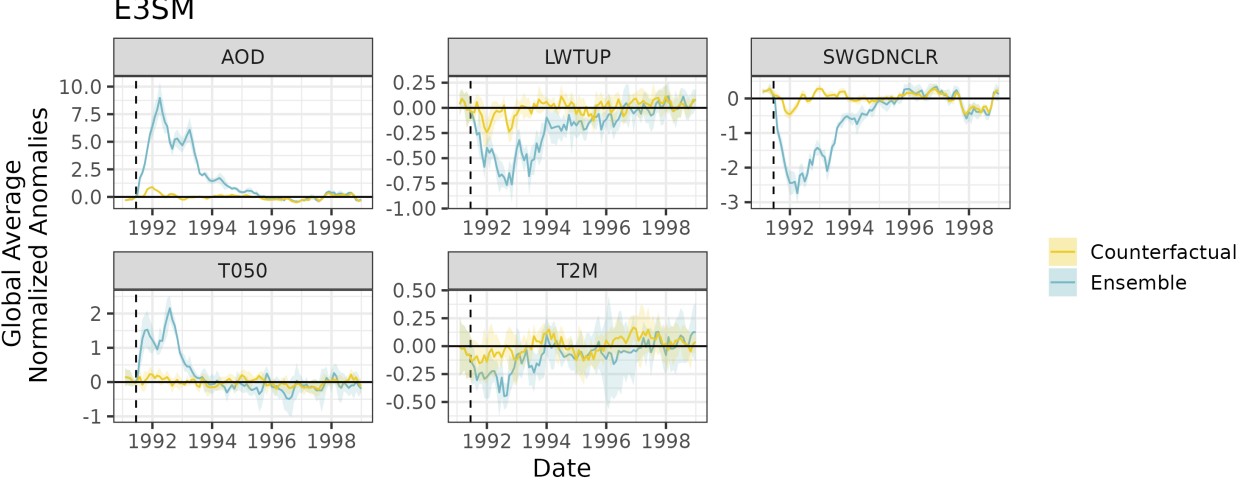

**Figure 6.** Globally averaged normalized anomalies for E3SM ensemble and counterfactuals. Note the y-axis is different for each plot. Shaded area represents ± one standard deviation of ensemble variability. Vertical dashed lines denote the June 15, 1991 Mount Pinatubo eruption.

### 3.2.2 Feature Importance

As with the HSW++ data, we performed a hyperparameter optimization for the E3SM data. Details on the hyperparameter used
and tuning is in Appendix A. After hyperparameter optimization, we used data from 1991-1998 to train the EESN to E3SM data
and compute stZFI. Since E3SM is a high-fidelity climate model, the data it produces will be a more realistic representation of
reality than HSW++. The pathways stZFI needs to quantify will be more complex and involve multiple variables.

Figure 7 shows stZFI for the E3SM ensembles and their counterfactuals. The vertical dashed line represents Mount Pinatubo's
eruption. Although the temperature pathway in E3SM is not as direct as HSW++, containing interactions and confounding vari-
ables, the feature importance results tell the same story. For both the E3SM Stratospheric (T050) and Surface (T2M) models,
AOD's feature importance immediately spikes at the eruption and is relatively large compared to other variables, then tapers off
as time progresses. The importance of LWTUP remains relatively flat. Importance of SWGDNCLR does see a small increase
after the eruption, although it is within the bounds of the counterfactual stZFI. Although we expected radiation to have larger
stZFI values, it could be because radiation has a lower signal compared to AOD and its impact is largely captured through
AOD.

Much like HSW++, T050 is important for predicting itself after the eruption, while T2M is not. There is a spike in T2M
feature importance around 1996, but this is largely because one of the ensemble members has a dramatic increase in T2M at
that time (as seen by the large variation in Figure 6). There are no major trends in stZFI for the counterfactuals, but all variables
retain some degree of importance since the variables on the respective pathways affect T050 and T2M with or without an
eruption. A broader assessment of stZFI robustness using E3SM is provided in Appendix C.





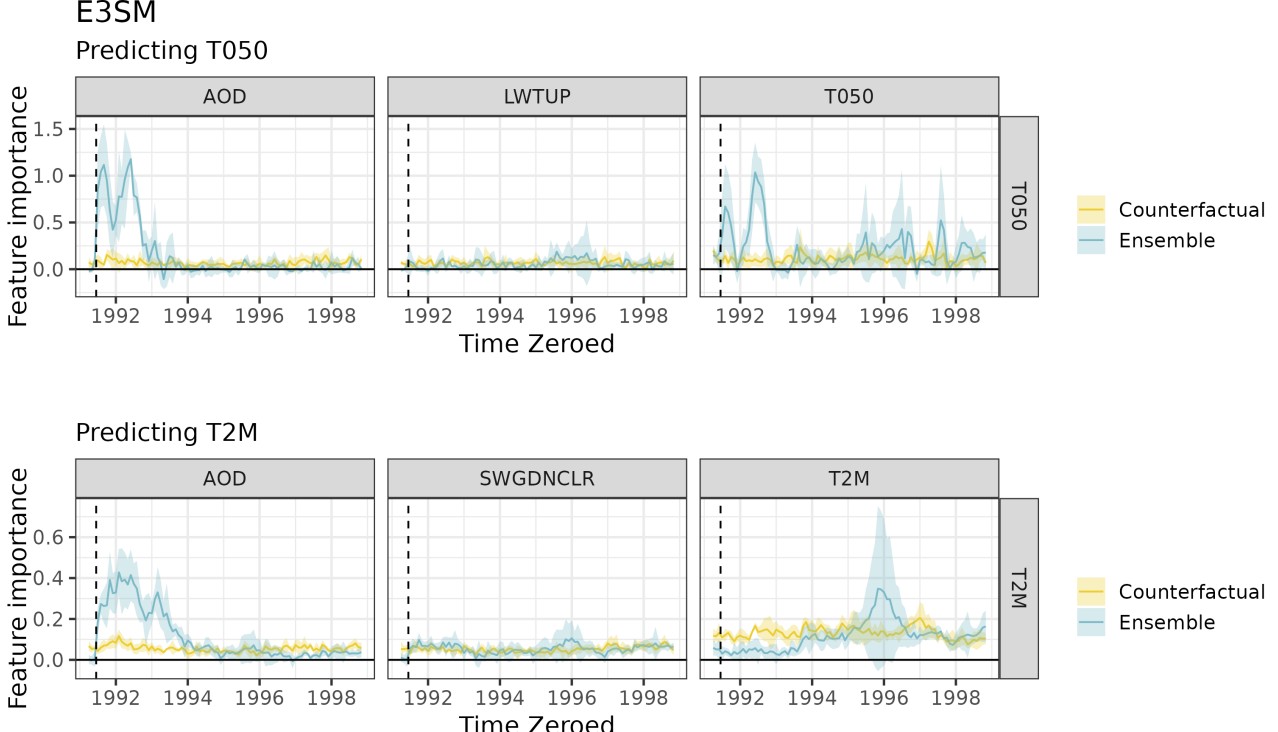

**Figure 7.** stZFI for E3SM ensembles and counterfactuals. Shaded region denotes ± one standard deviation of ensemble variability. Vertical dashed lines denote the June 15, 1991 Mount Pinatubo eruption.

Much like in the HSW++ case, we are able to extract relevant variable relationships from a purely data-driven ML model. As an EDA tool, this gives us an assessment of the relationships in the data. Unlike simple summaries such as means and correlations, the relationships found here are measured over time and account for complex, nonlinear associations. The latitudinal contributions to stZFI for the E3SM are deferred until Section 4.

### 3.3 MERRA-2 Application

The last two subsections applied stZFI to ESM simulations. Now we turn to verifying that the feature importance results from the ESMs are consistent with results from observed data products. To do this, we use Modern-Era Retrospective Analysis for Research and Applications, Version 2 (MERRA-2) (Gelaro et al., 2017) reanalysis as our observational data. As with the E3SM models, we build two EESN models for modeling temperature pathways with MERRA-2.

– *MERRA-2 Stratosphere (T050)* Model: Predict T050 given AOD, long-wave radiative flux net top of atmosphere (LWTUP) and T050.





– *MERRA-2 Surface (T2M) Model*: Predict T2M (two meter surface temperature) given AOD, short-wave radiative flux clear sky (SWGDNCLR) and T2M.

All input variables are time lagged. Vertically integrated AOD is taken from the variable TOTEXTTAU (Modeling et al., 2015a). We consider the years of 1991-1998 using monthly data, which provides climate information before and after the eruption of Mount Pinatubo. The spatial resolution is $1° \times 1°$ on a structured latitude/longitude grid.

### 3.3.1 Normalized Anomalies

Since MERRA-2 does not have a counterfactual, normalized anomalies are calculated differently for MERRA-2 than both HSW++ and E3SM. The normalized anomaly for each MERRA-2 variable $k$ at time $t$ and location $\mathbf{s}_i$ is calculated by:

$$Z_{k,t}(\mathbf{s}_i) = \frac{Z_{k,t}^O(\mathbf{s}_i) - \bar{Z}_{k,month(t)}(\mathbf{s}_i)}{sd(\mathbf{Z}_{k,month(t)}(\mathbf{s}_i))}, \tag{20}$$

where $\bar{Z}_{k,month(t)}(\mathbf{s}_i)$ and $sd(\mathbf{Z}_{k,month(t)}(\mathbf{s}_i))$ are the mean and standard deviation, respectively, across all data from 1991-1998 for the month in which time $t$ belongs (i.e. $month(t)$ returns the month of time $t$), for variable $k$ at location $\mathbf{s}_i$, $i = 1, \ldots, N$. Similar to E3SM, $k = 1$ refers to AOD, $k = 2$ refers to radiative flux (LWTUP for T050, SWGDNCLR for T2M) and $k = 3$ refers to temperature, T050 or T2M depending on which model is being discussed. Figure 8 shows latitudinal means of normalized anomalies over time from a single ensemble of E3SM around this time period. Figure 9 shows globally averaged normalized anomalies for MERRA-2. AOD and shortwave radiation have the largest relative spikes post-Pinatubo, while T2M and longwave radiation see the smallest change. The impact of Mount Pinatubo is clear on the shortwave radiation measurements and T050.

### 3.3.2 Feature Importance

After hyperparameter optimization, we used data from 1991-1998 to train the EESNs on MERRA-2 data and compute stZFI. Figure 10 shows stZFI for the MERRA-2 data, where the black vertical dashed line denotes Mount Pinatubo's eruption. There is a clear signal in the stZFI for AOD immediately after the eruption for both models, although the signal is clearer in the T050 model. Short-wave radiation also has a clear increase in importance after the eruption for the T2M model corresponding with the decrease seen in Figure 9. The importance for T050 predicting itself is more noisy, although it is elevated immediately post-Pinatubo eruption. The importance for T2M predicting itself shows a steady increase from 1991-1996, with a slight dip in 1995. This could potentially be due to an increase in auto-correlation of T2M post-Pinatubo. The importance of long-wave radiation is relatively flat. Similar to E3SM, we believe the feature importance for radiation is largely flat due to a lower signal to noise ratio compared to AOD, coupled with its correlation with AOD.



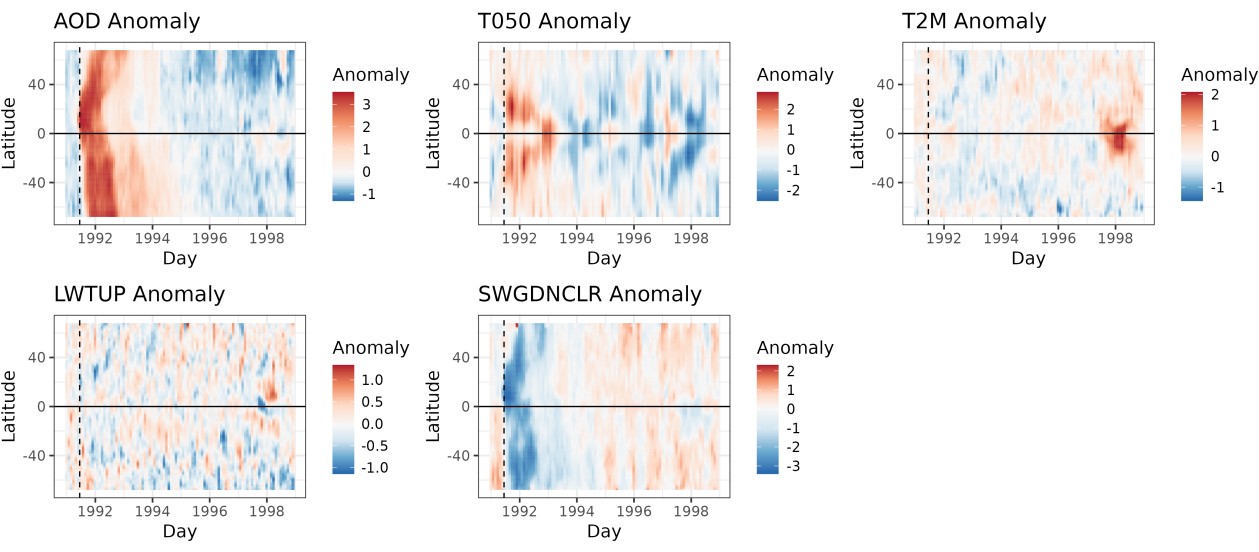

**Figure 8.** Latitudinal means over time for MERRA-2 normalized anomalies. Vertical dashed lines denote the June 15, 1991 Mount Pinatubo eruption.

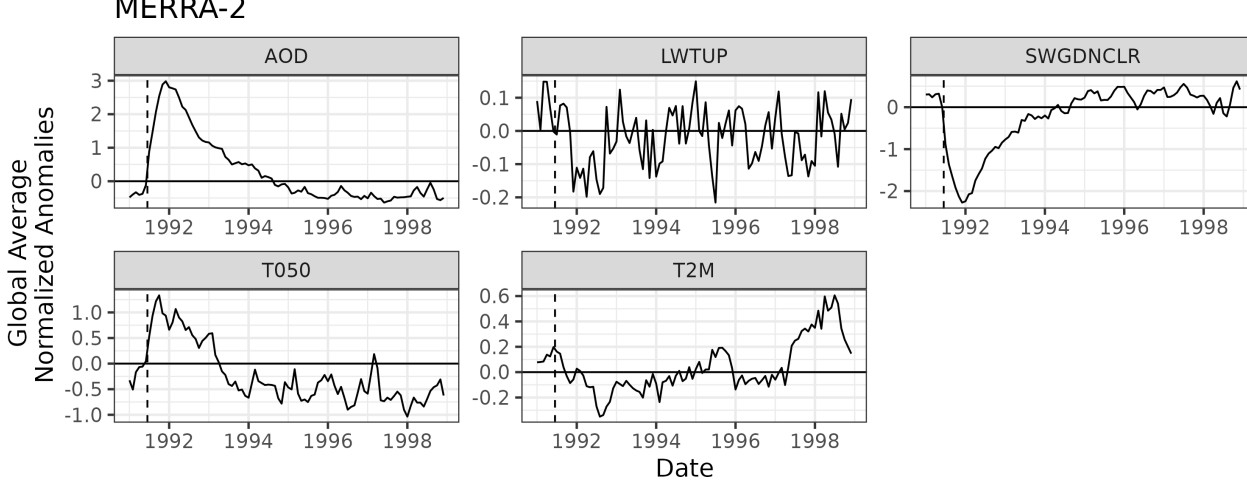

**Figure 9.** Globally averaged normalized anomalies for MERRA-2. Note the y-axis is different for each plot. Vertical dashed lines denote the June 15, 1991 Mount Pinatubo eruption.



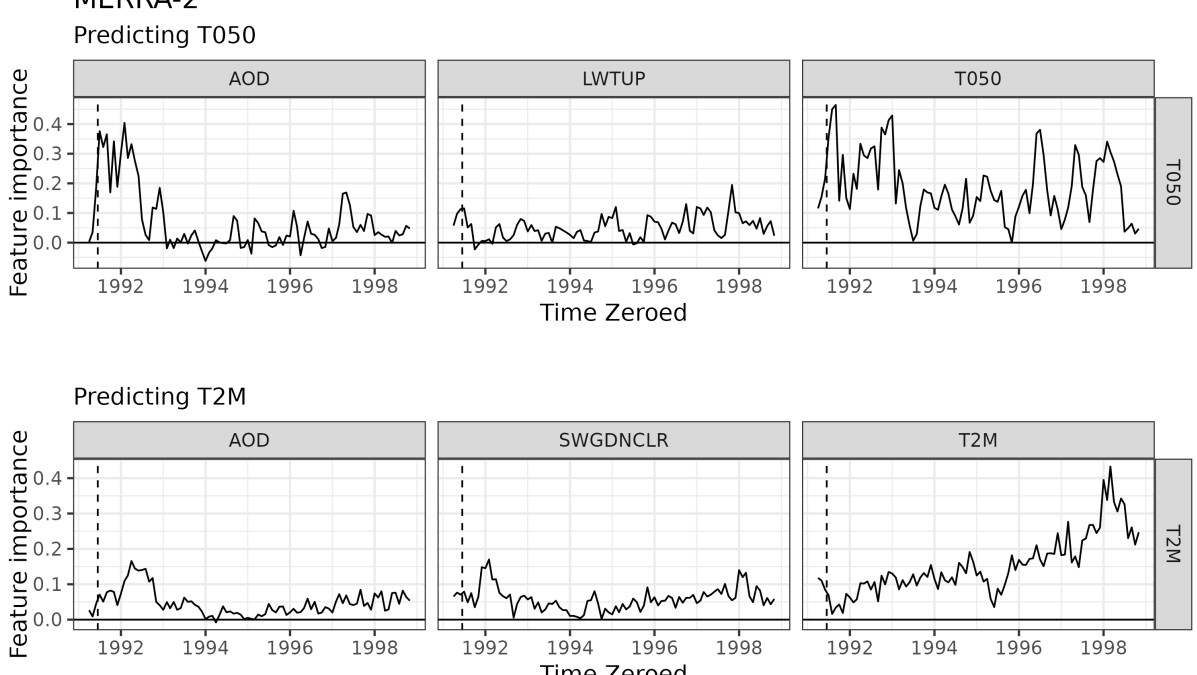

**Figure 10.** stZFI for MERRA-2. Vertical dashed lines denote the June 15, 1991 Mount Pinatubo eruption.

## 4 Comparing stZFI from E3SM to MERRA-2

The models for E3SM and MERRA-2 in Sections 3.2 and 3.3, respectively, are trained on the same time frame and spatial scale. We cannot compare observations to HSW++ since it is a notional lower fidelity model. Because the data are standardized in different ways, we will avoid exact quantitative comparisons, and make a qualitative comparison.

Considering effects globally averages over regional effects, so we consider the latitudinal contributions to stZFI to explore importance in both space and time. Figure 11 shows the *regional* contributions to stZFI for models predicting T050 using

E3SM and MERRA-2. These contributions show the relative importance of each of the three variables on predicting T050, by latitude, over time, providing a spatio-temporal feature importance. E3SM shows relatively uniform importance for AOD between -20°S and 30°N, from July 1991 to November 1993, with a slight decline in the winter of 1992. MERRA-2 shows importance for AOD in the regions -30°S to -10°S, and 10°N to 30°N, beginning in July 1991 and ending in the late summer of 1992. The FI for MERRA-2 appears to be more drawn out for T050, this could be due to model misspecification in E3SM

that does not fully account for the effects post-Pinatubo. E3SM and MERRA-2 largely agree on the importance of time-lagged T050 for predicting T050, as both show high importance near the equator at similar times. Trends for long-wave radiation are more difficult to assess, but it does appear that both E3SM and MERRA-2 have higher variability of FI near the equator, while towards the poles FI tends to remain more consistent. There is negative importance for longwave radiation immediately after





Pinatubo in both E3SM and MERRA-2, indicating its presence hurts the model. This likely due to long-wave radiation being
only weakly correlated with T050 combined with a relatively high noise compared to signal post-Pinatubo as shown in Figure
6. There is also a negative spike for AOD for E3SM in mid-1993, which could be due to AOD levels converging to those of the
counterfactual (Figure 6). Note that E3SM importances appear to be "smoother" since they are averaged over five ensembles,
whereas MERRA-2 is a single dataset.

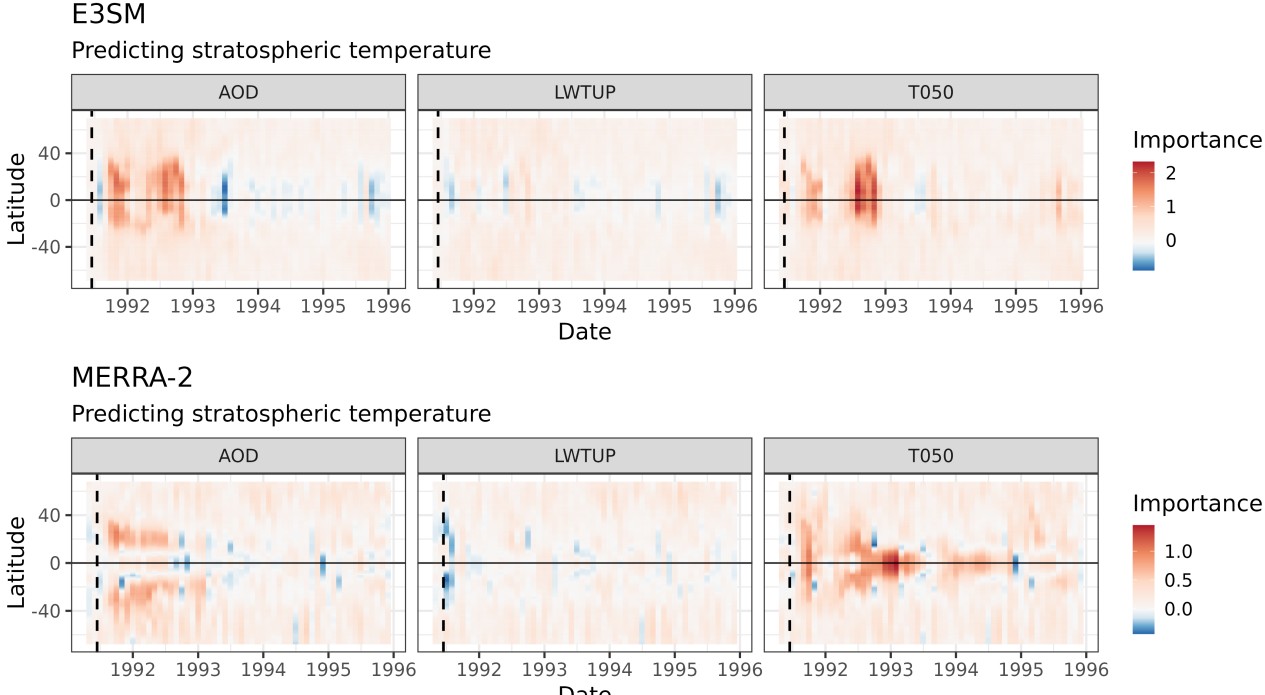

**Figure 11.** Latitudinal contributions to stZFI for E3SM and MERRA-2 for models predicting T050. Note importance scales are different for
E3SM and MERRA-2. Vertical dashed lines denote the June 15, 1991 Mount Pinatubo eruption.

Figure 12 shows the *regional* contributions to stZFI for models predicting T2M using E3SM and MERRA-2. E3SM sees
importance for AOD mostly just north of the equator post-Pinatubo, while MERRA-2 sees importance further from the equator
in both directions. The importance of shortwave radiation is spread across all latitudes for E3SM, while MERRA-2 has higher
importances further from the equator. There are not clear trends for importance of T2M, except during later years. E3SM puts
high importance on T2M on the equator at the end of 1995, while MERRA-2 shows importance at the end of 1994 just north
of the equator.



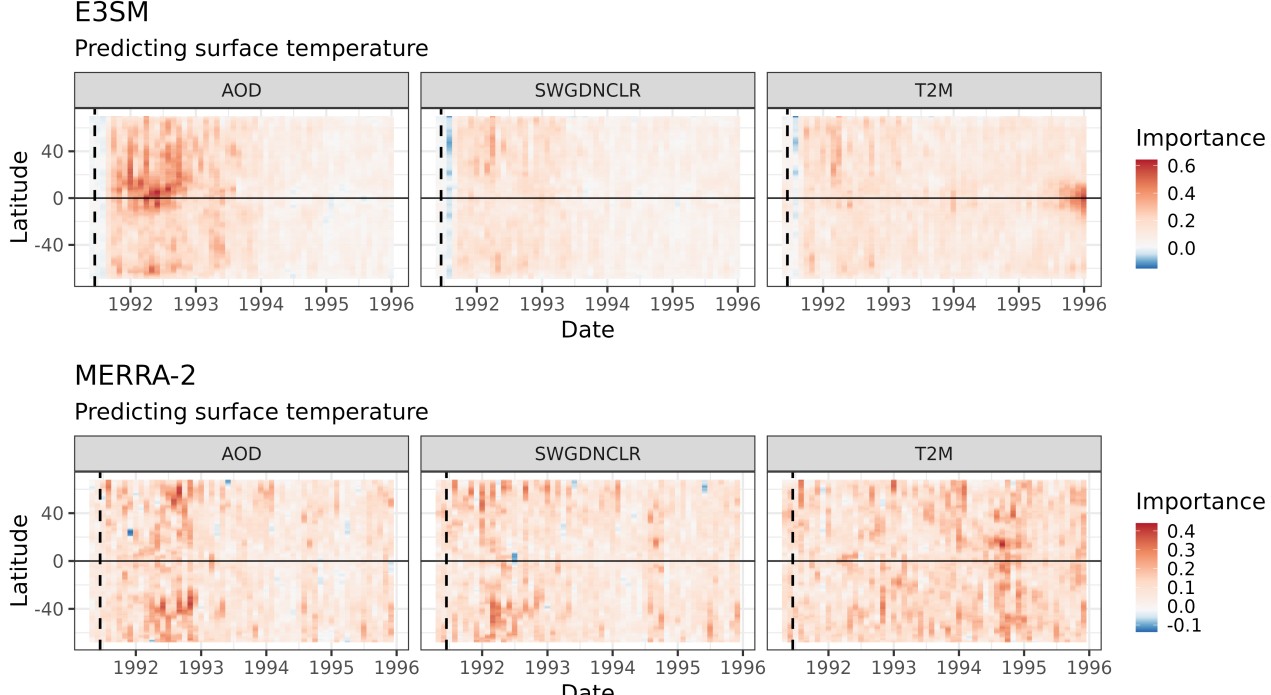

**Figure 12.** Latitudinal contributions to stZFI for E3SM and MERRA-2 for models predicting T2M. Note importance scales are different for E3SM and MERRA-2. Vertical dashed lines denote the June 15, 1991 Mount Pinatubo eruption.

## 5   Discussion

ESMs provide rich information about the physical state of the climate and its variations. ML and other data-driven models offer one path to taking advantage of the vast amounts of data produced by ESMs to advance the understanding of climate systems. In addition to using ML for predictive reasons, explainability methods allow for the ability to discover and quantify patterns in data via ML models for prediction. In this article, we provided an example of how an explainable ML technique, stZFI, can be used as an EDA tool for climate applications to understand how variable relationships evolve over space and time. We demonstrated stZFI via a case study that explored climate variable relationships associated with a natural exemplar of an SAI event: the 1991 volcanic eruption of Mount Pinatubo. We chose this event since it is well studied and documented, which helps future users understand how stZFI could be used as an EDA tool. We leveraged ESMs to study how stZFI quantifies variable relationships with datasets that are generated with known relationships. Further, we compared stZFI computed from ESM generated data to stZFI computed from reanalysis data to determine if the results were consistent.

We considered two climate pathways previously identified in the literature that are associated with the Mount Pinatubo eruption: (1) aerosols to long-wave radiative flux to stratospheric temperature changes and (2) aerosols to short-wave radiative flux to surface temperature changes. We applied stZFI to EESNs to conduct an EDA with an interest in understanding how the





three pathway variables are related to the changes in temperatures over time after an SAI. We studied these pathways using
three data sources: a simplified ESM with only the single forcing of aerosols (HSW++), a fully-coupled ESM (E3SM), and a
reanalysis dataset (MERRA-2). For all models and data sources, the relationships identified by stZFI were relatively consistent:

- *Aerosols* had the most consistent FI results. In all cases, there was a clear increase in stZFI for predicting temperatures
immediately after the SAI, which decreases over time.

- *Radiative flux* variables associated with E3SM and MERRA2 had relatively similar FI trends. For long-wave radiative
flux, there was no clear trend in FI with values close to 0 across all times when predicting stratospheric temperatures.
For short-wave radiative flux, there was a slight increase in FI after the SAI when predicting surface temperatures.

- *Temperature* FI values agreed between HSW++ and E3SM but differed from MERRA2 results. With stratospheric tem-
peratures, the HSW++ and E3SM results showed a clear increase in FI after the SAI, but the MERRA2 results showed
a noisy possible increase in FI. With surface temperatures, the HSW++ and E3SM results showed no FI trends, but the
MERRA2 results showed a steadily increasing trend in FI.

The consistency in FI results for AOD across data sources provides evidence of these variable relationships being a part of
the underlying mechanism. It is likely that AOD has the most consistent FI results due to the strong global signal to noise ratio
of AOD after the eruption in all data sources. That radiation and temperature FI do not agree could be partially due to E3SM
model discrepancies. These variables are changing at least partially due to the increase in AOD, making them downstream
effects of such an event. Additionally, it is possible that measurements are similar between MERRA-2 and E3SM while the
relationships that caused them could vary, manifesting itself in FI. The EESN itself is a relatively flat predictive model, meaning
it will likely not be able to capture all the complex relationships that exist, especially if they do not lead to better predictions.

The stZFI results can be used to point to new hypotheses and research directions. For example, the upward trend in stZFI
for T2M is unlikely due to Mount Pinatubo alone, and could lead to additional research. Another example suggested by the
latitudinal contribution plots is the question of how the latitude of an SAI event will affect its impacts. It also could help find
areas where climate models do not match observational data. stZFI shows the variables a model is using, and when, in order to
predict. Therefore, discrepancies between a climate model and observational data could point modelers to relationships a ESM
is not currently capturing.

In addition to using the SAI case study to demonstrate the ability of stZFI as an EDA tool, the analyses in this article
contribute towards an increased understanding and confidence that stZFI will return an accurate and reliable result. ESMs
played a key role in this process since they provide a specified cause with a known outcome, with a AOD being a major driver
in temperature changes both at the surface and in the stratosphere. In particular, this importance largely came from equatorial
regions, leaning slightly to the northern hemisphere. The ESMs also allowed us to examine results from counterfactual runs
where the SAI is removed. When we considered EESNs trained on the counterfactuals, we found no FI patterns associated
with the SAI. This result suggests that the FI trends that appear when SAI is included in the ESM runs are associated with
SAI and not some other phenomenon. With observational data only, there would be no way to know for sure whether feature
importance produced the correct effect, since this effect would not be known.



However, our analyses serve only as a case study for the assessment of stZFI. A more comprehensive evaluation of the method should be performed to better understand its strengths and limitations. For example, the literature on applying explainable ML methods to climate applications has been growing recently. A comparison of stZFI to other methods would be useful for understanding when stZFI is preferable to other methods. Additionally, the current methodology for stZFI does not fully account for correlation between input variables. Previous research suggests that ignoring correlation between variables can result in biased feature importance (Hooker et al., 2021). Further studies could be done to assess the affect of correlation on stZFI, and the methodology could be adjusted to better account for correlations. This future work would allow users to make stronger conclusions using stZFI without worrying about biases due to correlation.

Another direction for future research is developing a tool for ML EDA that is able to account for more complicated variable relationship structures. stZFI already provides an advantage over simple summary statistics such as means and correlations since it is applied to EESNs, which are flexible and not constrained to be linear or even monotonic. However, an EESN assumes a simple input-output model structure. We know the climate pathways, such as the ones we explore in this article, are more complicated. For example, the temperature pathway exploring the effects from an SAI event, to changes in radiation, to changes in temperature happen across multiple mechanisms. For example, the changes could vary depending on major climate cycles like ENSO, or such an event could possible affect ENSO, making it difficult to disentangle causes and effects. The EESN treats it as a simple prediction problem with temperature as the output and all other variables as inputs. A method that allows for more structure in the inputs, including interactions, could result in a more useful and representative explainability metric.

In this article, we presented stZFI as a tool for exploratory analyses. stZFI provides insights into climate events by quantifying variable relationships over space and time, which provides some insight into underlying mechanistic relationships. Exploratory analyses are an important aspect of science where new discoveries are made and hypotheses are generated. An additional objective in the climate science community is attribution (Hegerl et al., 2010; Bindoff et al., 2013). Although showing attribution is a multi-step problem, we believe stZFI could be used to provide an initial step towards making attribution claims. Regardless, we hope stZFI inspires ideas for how ML could be used for attribution in the climate space.

**Code and data availability**

The HSW++ code and data is available in Hollowed et al. (2024a). E3SM data and code is available in https://zenodo.org/records/12169924 (Ries et al., 2024). MERRA-2 data is publicly available (Modeling et al., 2015a, c, b).





## Appendix A: EESN Hyperparameter Details

This appendix provides details on the EESN hyperparameter tuning and selection for the models applied to HSW++, E3SM, and MERRA-2. We select hyperparameters by performing a hyperparameter search over the grid of values: $n_h = \{25, 50, 100, 200\}$, $U_{width} = \{0.1, 0.5\}$, $W_{width} = \{0.1, 0.5\}$, $U_\pi = \{0.1, 0.5\}$, $W_\pi = \{0.1, 0.5\}$, $\nu = \{0.1, 0.5\}$, $\lambda_r = \{.5, 5, 50\}$, where data is split into training and testing sets. HSW++ used days 0-800 as training, E3SM used dates 01-01-1991 to 12-31-1994 as training, and MERRA-2 used dates 01-01-1991 to 12-31-1994 as training. The prediction metric optimized for was root mean squared error (RMSE). The remaining times in each dataset were used for testing. We opted to use 20 EOFs for each variable for each of the data sets for consistency. For HSW++, 20 EOFs represents 98%, 82%, and 60% of the variation in AOD, T050, and T1000, respectively. For E3SM, 20 EOFs represents 93%, 92%, and 73% of the variation in AOD, T050, and T1000, respectively. For MERRA-2, 20 EOFs represents 88%, 87%, and 56% of the variation in AOD, T050, and T1000, respectively. We acknowledge the number of EOFs could differ by model and by variable within a model. Table A1 shows the optimal hyperparameters for each model for each dataset based on the hyperparameter search.

| Data | Model | $n_h$ | $U_{width}$ | $W_{width}$ | $U_\pi$ | $W_\pi$ | $\nu$ | $\lambda_r = 5$ |
|---|---|---|---|---|---|---|---|---|
| HSW++ | T050 | 200 | 0.5 | 0.1 | 0.1 | 0.5 | 0.1 | 50 |
| HSW++ | T1000 | 100 | 0.1 | 0.1 | 0.5 | 0.5 | 0.1 | 50 |
| E3SM | T050 | 50 | 0.1 | 0.1 | 0.5 | 0.5 | 0.1 | 5 |
| E3SM | T2M | 200 | 0.1 | 0.1 | 0.1 | 0.1 | 0.1 | 5 |
| MERRA-2 | T050 | 200 | 0.1 | 0.1 | 0.5 | 0.5 | 0.1 | 50 |
| MERRA-2 | T2M | 200 | 0.1 | 0.1 | 0.1 | 0.1 | 0.1 | 5 |

**Table A1.** Hyperparameters used for EESN models based on lowest test set RMSEs in hyperparameter search.

## Appendix B: Predictive Performance of EESN on HSW++ and E3SM

Figure B1 shows the predictive performance of the EESN over time for both HSW++ models. The three rows in each plot show globally weighted RMSE using different training sets; for example, the first row corresponds to training using data using times 1-200, then testing on data from 201-1200. Weights are calculated by taking the square root of the cosine latitude (Huth, 2006). That is, the weight associated with location $\mathbf{s_i}$ is

$$w_{\mathbf{s}_i} = \sqrt{cos\left(latitude(\mathbf{s}_i) \times \frac{\pi}{180}\right)}, \tag{B1}$$

where $latitude(\mathbf{s}_i)$ returns the latitude of location $\mathbf{s}_i$ in degrees. The EESN RMSE is compared to *replicate RMSE*, which uses four E3SM ensembles' values as predictions for the remaining member, then averages over the RMSEs of the four. This process is repeated five times (each ensemble is predicted using the other four), and the *replicate RMSE* is the average over those five results. All averages are calculated on a month by month basis. The EESN's RMSEs are typically lower than the replicate



RMSE, showing the EESN has lower prediction error than the natural variability of the climate system itself. This shows that the EESN is providing predictive ability beyond that due to ensemble variation, which, combined with the hyperparameter optimization, provides credibility to stZFI computed from this EESN. Figure B1 shows the EESN is able to capture trends better than that due to ensemble variability when given enough training data.

## Predicting stratospheric temperature

## Predicting surface temperature

**Figure B1.** Time series cross-validation global weighted average RMSE for both HSW++ models. Models are trained through the time shown on the row label. Bold blue lines are the average RMSE for an EESN, and light blue lines are the individual ensembles' RMSEs. The yellow lines represent the *replicate RMSE* used as a baseline comparison.

Figure B2 shows the predictive performance of the EESN over time using globally weighted root mean squared error (RMSE). The three rows in each plot show globally weighted RMSE using different training sets; for example, the first row

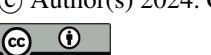

corresponds to training using data from 1991-1993, then testing on data from 1994-1998. This shows the EESN is able to capture trends better than that due to ensemble variability when given enough training data.

**Figure B2.** Time series cross-validation global weighted average RMSE for both E3SM models. Models are trained through the row label year. Bold blue lines are the average RMSE for an EESN, and light blue lines are the individual ensembles' RMSEs. The yellow lines represent the *replicate RMSE* used as a baseline comparison.

Figure B3 shows the predictive performance of the EESN over time using RMSE. The three rows in each plot show globally weighted RMSE using different training sets; for example, the first row corresponds to training using data from 1991-1993, then testing on data from 1994-1998. This shows the EESN is able to capture trends better than that due to ensemble variability

when given enough training data.





**Figure B3.** Time series cross-validation global RMSE for both MERRA-2 models. Models are trained through the row label year.

## Appendix C:  Assessing stZFI Robustness on E3SM

Assessing the robustness is important to understanding the behavior of a method. Here we present several checks (non-exhaustive) to help illustrate how stZFI behaves under different model specifications. All EESNs in this section were trained with the same hyperparameters. For the T050 model, $n_h = 50, \nu = 0.1, U_{width} = 0.1, W_{width} = 0.1, U_\pi = 0.5, W_\pi = 0.5, \lambda_r = 5$. For the T2M model $n_h = 200, \nu = 0.1, U_{width} = 0.1, W_{width} = 0.1, U_\pi = 0.1, W_\pi = 0.1, \lambda_r = 5$. These are the optimized hyperparameters used for the E3SM model in the main paper.





## C1 Prescribed Variation

We consider the impacts of eruptions smaller and larger than Mount Pinatubo to measure the gradient of the effects. These simulations are shorter, going from 1991-1995 and are initialized using historical CMIP6 ensemble members. For these pre-
scribed ensembles, we consider eruptions of 0.0x, 0.5x, 1.0x, and 1.5x the Mount Pinatubo eruption. The 0.0x eruption is the counterfactual and excludes both the Mount Pinatubo and Cerro Hudson eruptions. Five ensembles are generated for each eruption mass condition.

    Figure C1 shows globally averaged normalized anomalies for the prescribed variation E3SM ensembles. The colors correspond to the size of the prescribed eruption relative to Mount Pinatubo (e.g. 0.5 means the simulation replaces the original
Mount Pinatubo eruption with an eruption half the size.) There is a clear gradient in variable value corresponding to the size of eruption for all variables. The two peaks in AOD result from the standardization process; the variation in AOD for the early months of 1993 was low relative to its mean deviation from the counterfactual.

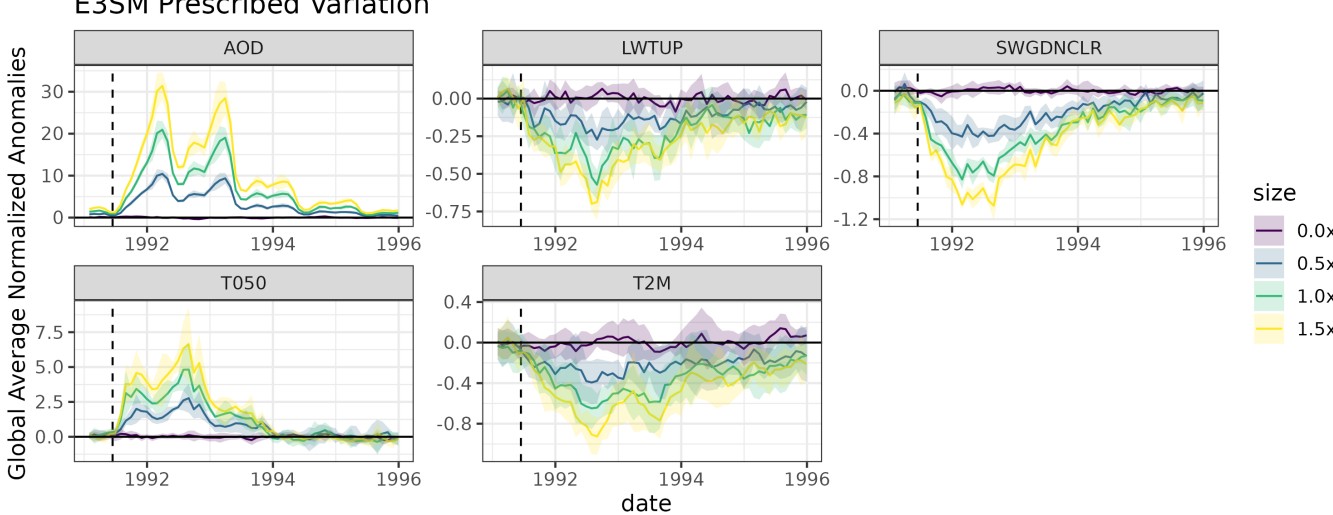

**Figure C1.** Globally averaged normalized anomalies for prescribed variation E3SM. Note the y-axis scale is different for each plot. Shaded area represents ± one standard deviation of ensemble variability.

    Figure C2 shows stZFI for E3SM with prescribed eruptions. The spike in importance is relative to the magnitude of the eruption: larger eruptions have larger spikes in feature importance. The importance of LWTUP and SWGDNCLR is minimal,
and T050 and T2M follow similar trends to the results in Figure 7.

## C2 Adding White Noise Variable

To better convince ourselves stZFI does not pick up on unimportant signals, we fit an EESN with an additional variable that is simulated from a standard Gaussian across all time and locations; denote this variable as white noise (WN). Figure C3 shows





## E3SM Prescribed Variation

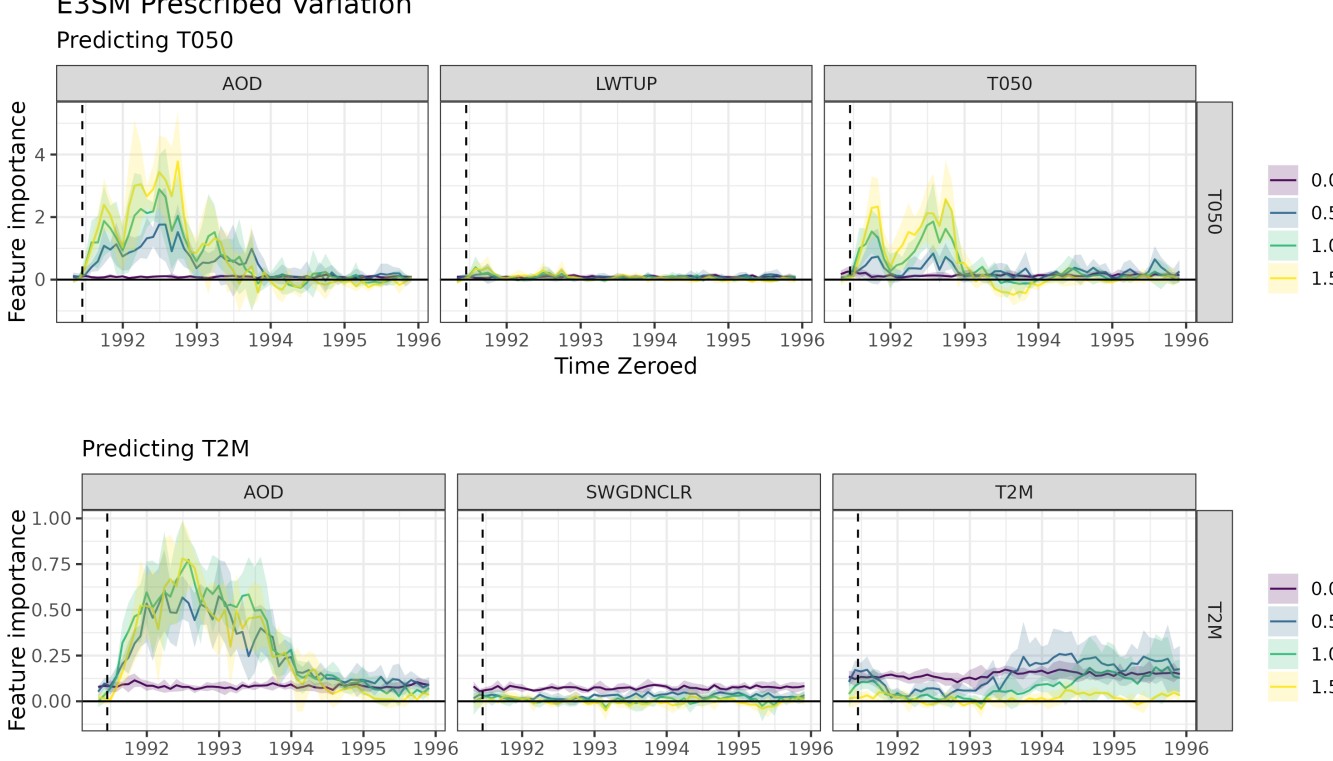

**Figure C2.** stZFI for E3SM simulations with prescribed eruptions. Color denotes relative size of eruption compared to actual Pinatubo eruption. Note the y-axis scales are different for the two rows.

stZFI for the E3SM ensembles with WN added as an input, for the T050 and T2M models. The feature importance for WN hovers close to zero for both models at all times. This result suggests that stZFI is not erroneously thinking it is important for predictions.

### C3    Adding Higher Signal to Noise Variables

Figure C4 shows globally averaged normalized anomalies for for two additional variables from E3SM: AODSO$_4$, which is integrated sulfate aerosol extinction coefficient (absorption + scattering, m-1) at 0.55 $\mu m$ wavelengths through the entire atmosphere, and BURDENSO$_4$, which is the column burden mass of sulfate aerosol. Both of these have a cleaner signal due to Mount Pinatubo than AOD (compare signal to noise of these variables to in Figure 6).

Figure C5 shows stZFI for the E3SM ensembles with extra variables for the T050 and T2M models. Considering the relative ranking of signal sizes seen in Figure C4, the magnitudes of stZFI in Figure C5 are consistent, with variables that change more after the eruption *and* are related to the outcome have greater feature importance. This gives evidence to the idea that feature importance is a metric that looks at both changes in degree of association and changes in feature magnitude. Radiation FI sees



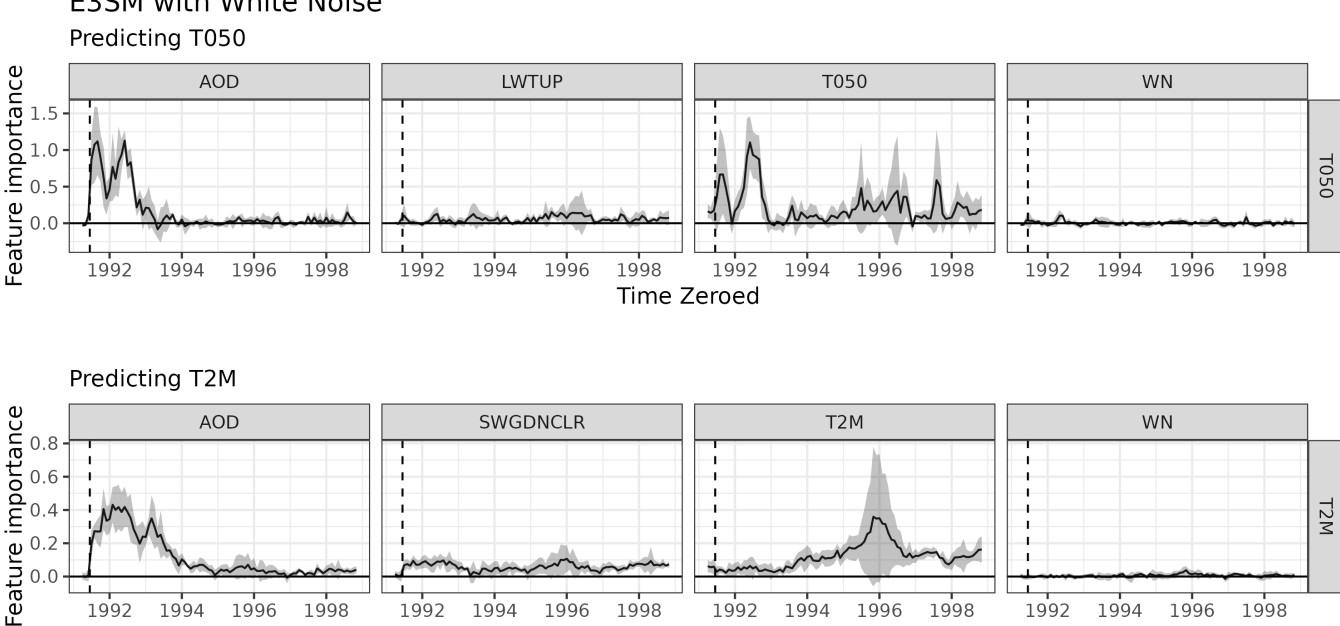

**Figure C3.** stZFI for E3SM with WN added as an input. Note y-axis scale differs for the two rows.

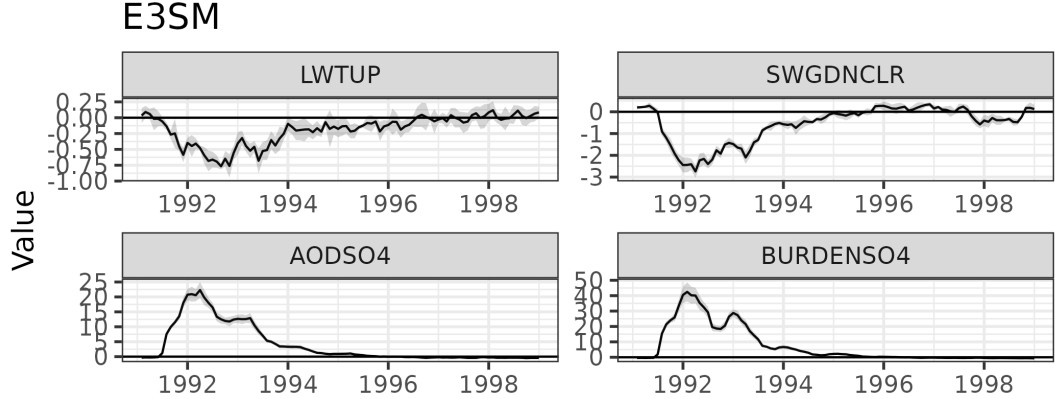

**Figure C4.** Globally averaged normalized anomalies for additional E3SM variables. Note the y-axis is different for each plot.

a slight decrease compared to the case without the additional variables; lagged temperature sees and even bigger attenuation. This points to the additional variables having a bigger impact. This is potentially due to collinearity between the variables.



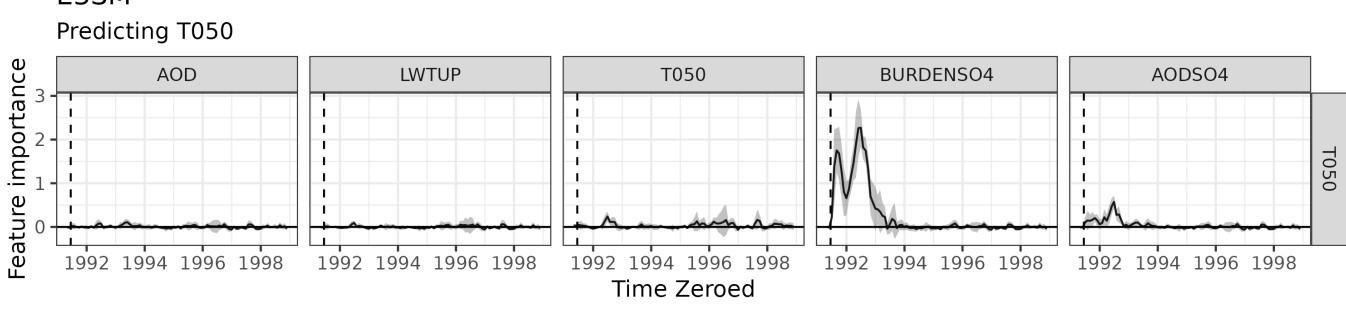

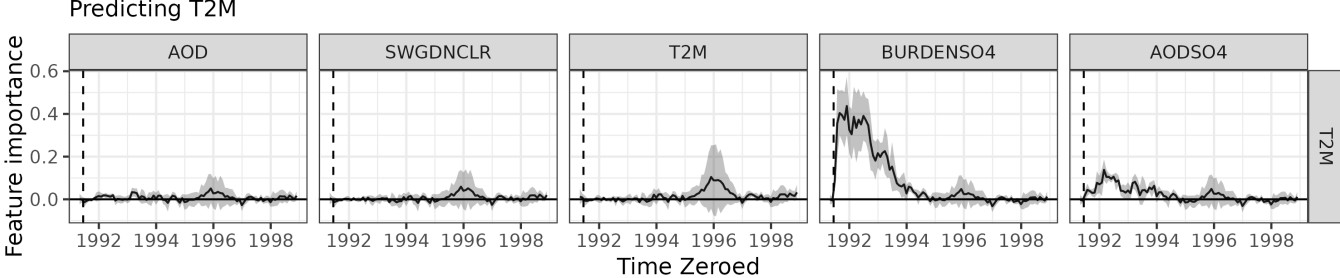

**Figure C5.** stZFI for EESNs fit using E3SM with additional variables that have higher signal-to-noise than AOD. Note y-axis scale differs for the two rows.

## C4 Excluding AOD from EESN

Figure C6 examines stZFI for a model that does not include AOD to a model that does include AOD. This moves in the opposite
direction of the previous section, where we remove an important variable instead of adding one to see the impact on FIs. This
will give us an idea of the collinearity of AOD with the remaining features. Trends between the two are mostly similar, except
that stZFI for the remaining variables tend to be higher in the model without AOD, likely meaning there is collinearity between
input variables, and without AOD, other variables account for that relationship.

*Author contributions.* DR, KG, KM developed the methodology. DR, KG, KM wrote code to implement stZFI. DR ran the analyses. BH
aided in the climatological interpretations. All authors helped reviewing the paper.

*Competing interests.* The authors declare no competing interests are present.



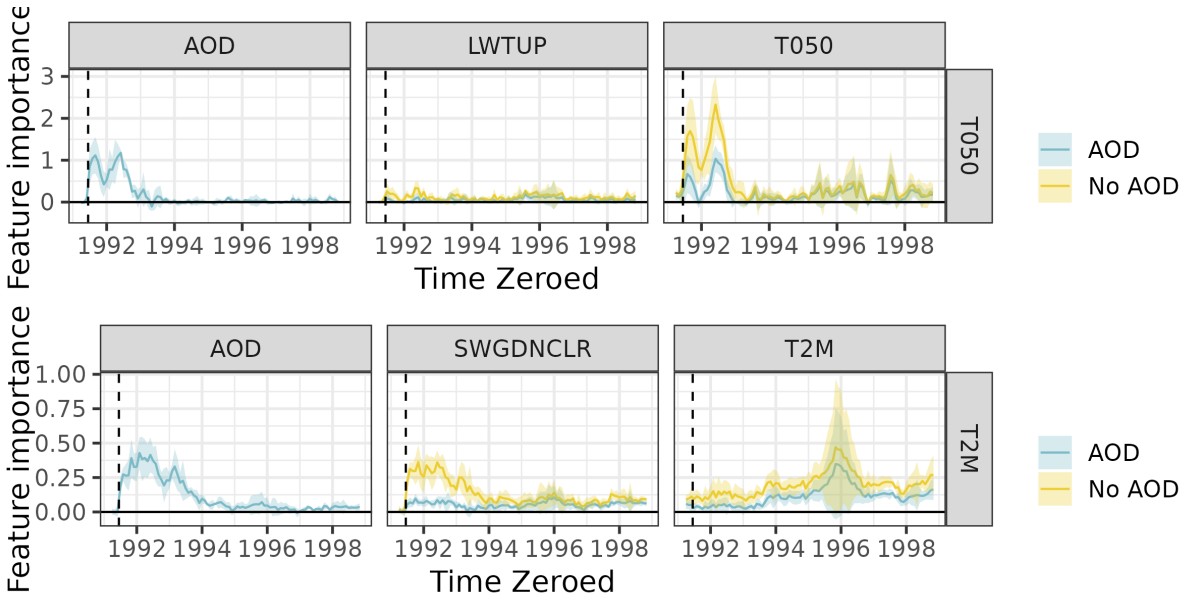

**Figure C6.** stZFI for E3SM when AOD is not included in EESN. Note y-axis scale differs for the two rows.

*Acknowledgements.* This work was supported by the Laboratory Directed Research and Development program at Sandia National Laboratories, a multimission laboratory managed and operated by National Technology and Engineering Solutions of Sandia LLC, a wholly owned subsidiary of Honeywell International Inc. for the U.S. Department of Energy's National Nuclear Security Administration under contract
DE-NA0003525. This paper describes objective technical results and analysis. Any subjective views or opinions that might be expressed in the paper do not necessarily represent the views of the U.S. Department of Energy or the United States Government. This research used resources of the National Energy Research Scientific Computing Center (NERSC), a Department of Energy Office of Science User Facility using NERSC award BER-ERCAP0026535. SANDXXXXXX.



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
