# Peer review of "Using feature importance as exploratory data analysis tool on earth system models"

_Geoscientific Model Development, 2024_

## Author Comment (AC1)

**Summary of responses**

We provide responses to reviewer 1's comments in this document. Reviewer comments are in normal text and in black. Our responses are in blue. We would like to thank both reviewers for time and expertise since their comments contributed to making this a better manuscript.

**1 Reviewer 1**

**1.1 General comments**

In this paper Ensemble Echo State Networks (EESNs) are built based on data from Earth System Models (ESMs) to investigate the machine learning explainability technique of spatio-temporal zeroed feature importance (stZFI) to see the effect of different variables on each other in case of a large stratosphere aerosol injection, in this paper the natural event of the volcanic eruption of Mount Pinatubo. The paper contains several experiments ranging from simpler models to reanalysis data, showing the potential of the method. Although the main idea of the paper is clear, the details can be hard to follow, partly due to lack of explanation, or by using not the correct term.

**1.2 Specific comments:**

- Line 44-45: What is a replicate hold out set vs a repeated hold out set? I think some more explanation would be good.
  *We will add the following after line 45: Replicate hold-out uses many independently generated full time series from the same ESM for the same period, with one series chosen as the training set and another as the testing set, while repeated hold-out splits single timer series into training and testing sets. Notably for replicate hold-out, the train and test set cover the same time span whereas in repeated hold-out the test set always covers the future relative to the training set.*

- Line 88: How do these temperature differences propagate over time?

  *We will add the following to line 88 to reference Figure 8, which shows latitudinal anomalies over time for T2M and T050 for MERRA-2: Figure 8 how T050 and T2M propagates over this time.*

- Line 94: When you are using reanalysis data, you are looking at a combination of models and observations, how would you get interesting relationships in only observations from this?

  *This was poor wording on our end. We shouldn't have used the word "observations" the second time. You are correct, our method cannot pull out differences in the reanalysis data due to the observed components and the modeled components. We will change this sentence to: We additionally consider one reanalysis dataset (i.e., combination of observed*

*and model data) to demonstrate the ability of stZFI to find interesting relationships and quantify how they evolve over time.*

- Line 206: Replace 'real' by 'reanalysis' Corrected

- Line 242-243: I am not sure I understand the difference between this paper and McClernon et al. (2024), would it not be better to also take T050 into account to forecast T1000?

  We were focused on the two specific climatic pathways in this manuscript, one corresponding to the surface and the other, the stratosphere. By keeping the temperatures in separate models we are able to model the mechanism based on the climatic pathways. In McClernon et al. 2024, they focused more on predictability, and because of that, added T050 when forecasting T1000 because it gave better predictions. But here we are more interested in using FI for EDA for inference and scientific exploration, as opposed to purely prediction. We will add the following at line 244: *We make the distinction here that we focus on distinct climatic pathways for surface and stratosphere in an attempt to isolate affects.*

- Line 246: Ensemble -¿ Ensemble member. The difference between ensemble and ensemble member is not clear throughout the paper.

  We have gone through the manuscript and updated "ensemble" and "ensemble member" to be consistent and clear to avoid confusion.

- Line 255-261: To be more clear, I think it would be good to guide the reader a bit more through Figure 1, why the temperatures in the NH have a negative anomaly for example, why the aerosol spread is much faster over the NH etc.

  We propose adding the following to the paragraph in lines 255-261: *The injection of aerosols at time $t = 179$ and how it propagates in space and time is seen in Figure 1 and 2. Both T050 and T1000 are directly related to AOD by construction in the HSW++ simulations, with increases in AOD contributing to longwave heating of the upper levels and shortwave cooling by increased scattering of incoming solar radiation in the lower levels. This radiative heating is parameterized in HSW++ by introducing temperature tendency terms parameterized by AOD, since HSW++ does not include an explicit radiative transfer model (see Hollowed et al. 2023). Increased stratospheric AOD results in a positive temperature anomaly in T050 due to thermal absorption, and a negative anomaly in T1000 due to increased scattering/reflection of shortwave radiation. AOD is advected by the stratosphere circulation faster in the northern hemisphere due to the fact that the injection occurs in the northern hemisphere, and the mean stratosphere circulation tends to be poleward. In the counterfactual, there is a small spike in T050 around day 270 that is unrelated to an aerosol injection (counterfactual AOD is zero).*

- Line 261: Could you explain why there is this unrelated spike?

  Because there is only a single counterfactual run, this spike is due to normal variation. If Hollowed et al. (2023) had done an ensemble of counterfactual runs, with perturbed initial conditions, and then computed the ensemble mean, this spike would probably go away in the limit of many ensemble members. The authors likely did not do this because the HSW++ counterfactual should have less variability relative to the the full atmosphere, and especially relative to the fully coupled runs. We will add the following to line 261: *This spike is due to normal variation. Had there been more than one counterfactual, taking the average would likely smooth over this effect.*

- Line 267: How much time-lag is there?

  $\tau = 1$. $\tau_{emb} = 1, m = 3$, as explained in Section 3 second paragraph.

- Line 269: I do not really see smoother decay for T050?

  This was a typo. We meant to say that because model T050 has higher peak importance for AOD, that it also has a steeper decay of importance compared to the T050 model. We will change this sentence to: *The importance for AOD is higher for the T050 model compared to T1000 model, and the decay of importance is steeper from its peak for the T050 model compared to the T1000 model.*

- Figure 4: Why is there a negative value for importance after the positive values for T050?

  Figure 4 is for a single ensemble member, so the magnitude of the negative feature importance is exaggerated compared to the ensemble mean, as seen in Figure 3. The fact that it goes negative is likely due to AOD levels tapering off and the relationship reversing trend.

- Line 326: I think it would be good to explain a bit more about Figure 5, what we are seeing and why. T2m for instance has a large positive anomaly around 1996, this is an outlier?

  We will add the following after line 326: *Anomalies by latitude band are computed based from Equation 19. Due to the Pinatubo eruption, we see large, positive anomalies in AOD and T050, with T050 largely focused around the equator in the years following the eruption. We also see smaller, negative anomalies for T2M around the equator and north of the equator. The large positive anomaly in T2M around 1996 is due to the spike in temperatures for ensemble member 1 (this heatmap is for ensemble member 1 only). Shortwave radiation also has negative anomalies after Pinatubo, particularly in the northern hemisphere, likely relating to the negative anomalies for T2M.*

- Line 347: Looking at Figure 6, there is no anomaly for T2m around 1996. Does Ensemble member 1 have that much influence on the feature importance?

Looking at Figure 6, you can see significant variation around 1996 for T2M. Ensemble 1 impacts the FI of Ensemble 1 only, but when we calculate variation from ensemble-to-ensemble of FI it will show up in variance like in Figure 7.Taking the mean over only 5 ensemble members will result in a skewed final result when there is a significant anomaly. The benefit of quantifying the variation is when we see as much variation as we do around T2M at 1996 is we know there is a lot of ensemble variation. In this case, one extreme ensemble member has a significant effect, and creating a larger ensemble to average over would likely drown out this variation. Having a larger ensemble to average over could alleviate this issue. We will add the following to line 347: *More than five runs of the ESM ensemble would be necessary to determine if this spike is truly anomalous or part of a larger trend.*

- Line 375: Figure 8 is about MERRA, not about E3SM?

  Corrected

- Line 377: There is again a large anomaly for T2m, this time around year 1998. Could you explain this further?

  This is likely due to the 1997-1998 El Nino, which was the largest recorded at that time, and led to a significant increase in globally averaged temperature. See `https://climate.nasa.gov/vital-signs/global-temperature/?intent=121` for the relative increase in T2M. We will add the following to line 377: *The large anomaly for T2M aroudn 1998 is likely to the 1997-1998 El Nino, which was the largest recorded at that time, and led to a significant increase in globally averaged temperature.*

- Line 449: Could you add some explanation about whether this increasing value for T2M is significant?

  This general increasing trend of importance for T2M also matches what we see in `https://climate.nasa.gov/vital-signs/global-temperature/?intent=121`, which could be a sign of general climate warming. In the reanalysis case, we only have a single set of data and therefore don't have a way to quantify the uncertainty to truly assess whether the increasing trend is significant. We will add the following to line 449: *It is possible this upward trend is due to a combination of increasing global surface temperatures and a strong El Niño event from May 1997 to May 1998 (Wang and Weisberg 2000).*

- Line 518-526: From the text is not clear which figures are for which models/reanalysis. The text about Figure B2 is equal to the text about Figure B3.

  Corrected.

**1.3 Technical**

- Line 86: Temperatures -¿ temperatures corrected

- Line 125: without -¿ Without Corrected.

---

## Author Comment (AC2)

**Summary of responses**

We provide responses to reviewer 2's comments in this document. Reviewer comments are in normal text and in black. Our responses are in blue. We would like to thank both reviewers for time and expertise since their comments contributed to making this a better manuscript.

**1 Reviewer 2**

**1.1 General comments:**

Ries et al. (2024) primarily apply the spatio-temporal zeroed feature importance (stZFI), an explainable AI (XAI) tool, to investigate the relationships between various variables associated with a stratospheric aerosol injection event. Notably, this stZFI method can reveal how the feature importance of predictors evolves over time. Utilizing this approach, the authors evaluate the time-variant contributions of volcanic aerosols to the prediction of local and surface temperature. They validate the results with mutiple datasets, including both model simulations and observations, demonstrating that stZFI can identify relationships consistently accross different datasets. This article showcases the capability of stZFI as an exploratory data analysis tool in climate research with great detail and precision. However, I would recommend the authors to devote more effort to explain the stZFI results physically. Please find my comments as below.

**1.2 Specific comments:**

- Line 9-10: The meaning of this sentence is unclear to me. Here the authors only use feature importance to distinguish between signal related to volcanic aerosols and others, not really natural climate variability.
  The goal of using multiple ensemble members is to be able to run the analysis and have some confidence that the signals that emerge in the ensemble vs CF are real, and not due to natural variability. I.e., if we'd just used a single ensemble member and single CF simulation, we wouldn't be able to say with confidence that any difference between those simulations are real, of if you just got the result you got because you picked a lucky (or unlucky) possible climate state by chance. *The use of perturbed initial condition ensembles introduces variability mimicking the natural variability in the atmosphere, thus the signals emerging using FI can be evaluated against the natural variability in the climate system.*

- Section 1.1: It is necessary to include the possible latitudinal transport of volcanic aerosols driven by the large-scale circulation in stratosphere - the Brewer-Dobson circulation (Butchart 2014).
  We agree that the BD circulation is largely responsible for the poleward transport of Pinatubo sulfate, and we don't believe we are saying anything

to the contrary. The focus of Section 1.1 is to explain the Pinatubo eruption in terms of its magnitude and motivate its use as an exemplar problem. We believe discussion of the BD circulation in detail is beyond the scope of this paper, which is focused on the development and application of a data-driven EDA method that leverages ESMs to gain insights into climate problems. We will add the following to the first paragraph of section 1.1 to make clear the effect BD circulation has on the transport of the Pinatubo sulfates: *The eruption released 18-19 Tg of sulfur dioxide into the atmosphere, causing changes to aerosol optical depth (AOD), transporting partially through the Brewer-Dobson circulation (Butchart 2014) and consequently changes to stratospheric temperatures (Sato et al., 1993; Guo et al., 2004)*

- Figure 1: The movement of aerosols from equator to polar regions could also be driven by the Brewer-Dobson circulation, not only just due to diffusion. Please clarify it.
  We will add the following to line 257:   *The injection and spread of aerosols due, in part to the Brewer-Dobson circulation, is clear in latitude and time.*

- Section 2.2.2: Why do you only consider latitude bands for regional contributions? Is there meridional transport of volcanic aerosols? If yes, it would be interesting to give a latitude-longitude global plot showing regional feature importance when T050/T2M peaks.
  We only show latitudinal bands for regional contributions since that is where we see the most variation in temperatures, both surface and stratospheric. We will add the following to line 194: *We focus on latitudinal bands since they account for the most variation in surface and stratospheric temperatures.*

- Line 233: What's the highest height of model outputs?
  Assuming the reviewer is asking about the model top: 0.1mb /  60 km (refer to the E3SMv2 overview paper, `https://agupubs.onlinelibrary.wiley.com/doi/10.1029/2022MS003156`). We will add the following to line 233: *Model outputs are remapped to a 2°× 2° structured latitude/longitude grid with 72 vertical levels up to 0.1mb /  60 km.*

- Figure 4: What does negative importance in three subfigures mean? When can people trust that the feature importance from stZFI is reflecting a real relationship?

  Negative feature importance implies the inclusion of the feature in question makes predictions worse than if it had not been included. Often, this will be due to overfitting. Because we are working with spatio-temporal features, it is possible and somewhat common to see feature importance go negative for a brief period of time and space such as in Figure 4. We will add the following to line 278: *Negative feature importance implies the inclusion of the feature in question makes predictions worse than if it*

*had not been included. However, small periods of negative stZFI is not a
concern, because it is a spatial-temporal metric so it is not unreasonable
to expect some time or spatial periods to not be helpful for prediction.*

- Line 439-440, Line 448-449: The T2M FI shows a large increase over
  1997/98 (Figure 10, subfigure for T2M). Could the increase of FI in this
  period be caused by the internal variability instead of the volcanic aerosol
  radiative effect? For example, there is a strong El Niño event from May
  1997 to May 1998 (Wang and Weisberg 2000), which could lead to higher
  autocorrelation in T2M.
  This assessment seems perfectly plausible. In line with a response to
  a similar comment from R1, we added the following to line 449: *It is
  possible this upward trend is due to a combination of increasing global
  surface temperatures and a strong El Niño event from May 1997 to May
  1998 (Wang and Weisberg 2000).*

---

## Author Response (AR2)

1/6/25

Author Response

1. We made grammatical corrections as pointed out by reviewer.
2. We added links to the data and code on github in section 3.2 and in the Code and Data Availability section as requested by editor.
3. We changed "HSW++" to "HSW-V v1.0" to match how it is refereed to in its published paper (Hollowed 2024b).